

# The role of geomorphology, rainfall and soil moisture in the occurrence of landslides triggered by 2018 Typhoon Mangkhut in the Philippines

Clàudia Abancó [1], Georgina L. Bennett [1], Adrian J. Matthews [2], Mark A. Matera[3], Fibor J. Tan[3]

[1]College of Life and Environmental Sciences, University of Exeter, Exeter, EX4 4RJ, United Kingdom
[2]Centre for Ocean and Atmospheric Sciences, School of Environmental Sciences and School of Mathematics, University of East Anglia, Norwich, NR4 7TJ, United Kingdom
[3] School of Civil, Environmental and Geological Engineering, Mapua University, Manila, Philippines

*Correspondence to*: Clàudia Abancó (c.abanco@exeter.ac.uk)

**Abstract**

*In 2018, Typhoon Mangkhut (locally known as Typhoon Ompong) triggered thousands of landslides in the area of Itogon (Philippines). A landslide inventory of 1101 landslides over a 570 km² area is used to study the geomorphological characteristics and land cover more prone to landsliding as well as the rainfall and soil moisture conditions that led to*

*widespread failure. Landslides mostly occurred in slopes covered by wooded grassland in clayey materials, predominantly facing East-Southeast. The analysis of both satellite rainfall (GPM IMERG) and soil moisture (SMAP-L4) finds that, in addition to rainfall from the typhoon, soil water content plays an important role in the triggering mechanism. Rainfall associated with Typhoon Mangkhut is compared with 33 high intensity rainfall events that did not trigger regional landslide events in 2018 and with previously published rainfall thresholds. Results show that: a) it was one of the most intense rainfall*

*events in the year but not the highest, and b) despite satellite data tending to underestimate intense rainfall, previous published regional and global thresholds are to be too low to discriminate between landslide triggering and non-triggering rainfall events. This work highlights the potential of satellite products for hazard assessment and early warning in areas of high landslide activity where ground-based data is scarce.*

## 1 Introduction

Landslides driven by typhoon and monsoon rainfall cause thousands of fatalities and millions of pesos in damage to infrastructure and commerce in the Philippines each year. The Philippines accounts for 46% of rainfall-triggered landslides in SE Asia, although it represents only 6% of the land area (Kirschbaum et al., 2015; Petley, 2012). The climate characteristics, with frequent tropical cyclones and two different monsoon regimes, together with abrupt orography and unstable geologic materials make the terrain prone to Multiple-Occurrence Regional Landslide Events (MORLEs) (Crozier, 2005). Despite the

relevance of the phenomena, the understanding of the triggering conditions and the instability mechanisms associated with



rainfall triggered MORLEs in the Philippines has still received little attention. This, combined with the incomplete landslide records and the shortage of landslide inventories, results in hazard and risk assessment techniques that still lack accuracy in the country.

The understanding of MORLEs and the assessment of their impact relies on the availability of landslide inventories (Crozier, 2005; Martino et al., 2020; Shu et al., 2019). Landslide inventories are key to evaluate the probability of slope failure, based on the conditions of previous slope failures and the effects of local terrain conditions across a region as a preliminary step toward landslide susceptibility, hazard and risk assessment (Fell et al., 2008; Guzzetti et al., 2005, 2012). Regardless of their importance, landslide inventories are often not available due to incomplete event records, or as a result of the lack of time and

resources to update them, for example in response to extreme events (Malamud et al., 2004). To map landslides across large regions using manual techniques is a highly time consuming task, and although the use of automatic mapping tools is increasing (Alvioli et al., 2018; Borghuis et al., 2007; Kirschbaum and Stanley, 2018; Scheip and Wegmann, 2020), their widespread applicability still presents some limitations. It is particularly challenging in regions hit by the passage of typhoons, where the area affected by landslides can be up to hundreds of $km^2$ (e.g.: Tseng et al., 2015).


In the Philippines, a nationwide inventory of >12000 landslides is available (Lagmay et al., 2017). However, most of the landslides are mapped as points rather than polygons, precluding magnitude-frequency analysis, a major component of landslide hazard assessment (Guzzetti et al., 2005). The use of polygon landslide inventories instead allows the retrieval of the landslide magnitude distribution (e.g.: Malamud et al., 2004; Parker et al., 2011). A limited number of studies including the

analysis of landslide conditioning factors in the area of Baguio have been published (Nolasco-Javier et al., 2015; Nolasco-Javier and Kumar, 2019), however the Philippines lacks a more detailed landslide susceptibility studies that may help in local planning. For example, the Philippines' Mines and Geosciences Bureau (MGB) hazard map for the area of Itogon (Benguet, Luzon) is based on extreme scenario and hence, classes most of the region at the highest hazard level, making land use planning difficult.


The analysis of the triggering rainfall conditions is also fundamental to understand MORLEs. The study of landslide triggering rainfalls has been of interest of the scientific community in recent decades, generating extensive literature. One of the most common approaches for the prediction of landslide triggering rainfalls is the definition of rainfall thresholds. Rainfall thresholds are used to characterise the rainfall conditions that, when reached or exceeded are likely to trigger one or more

landslides or torrential flows (De Vita et al., 1998). Different state of the art techniques and methodologies to obtain rainfall thresholds are reviewed by Segoni et al. (2018), while their applications for early warning purposes are assessed by Guzzetti et al. (2020). Two main approaches are used to derive such thresholds: a) physically-based models, where infiltration and hydrologic behaviour of the rainfall over a susceptible soil layer is simulated (e.g.: Crosta and Dal Negro, 2003; Godt et al., 2008; Papa et al., 2013); or b) empirically derived thresholds, based on the analysis of a database of rainfalls using, for example,



statistical techniques (e.g.: Brunetti et al., 2010; Guzzetti et al., 2007). The thresholds are generally expressed as a correlation between the peak intensity of the rainfall for different durations or the relationship between the total rainfall versus its duration (usually in the form $I=\alpha D^{-\beta}$), although some authors also include other triggering or antecedent rainfall parameters, as extensively reported by Segoni et al. (2018). Aspects such as the geographical distribution, the intended use of the thresholds, or simply the resources available determine the source of the rainfall data used to construct thresholds. Uncertainties on the

source and resolution of the data as well as on the methods used to define the rainfall events or the rainfall thresholds will be key to their accuracy (Abancó et al., 2016; Leonarduzzi and Molnar, 2020; Nikolopoulos et al., 2015). Rainfall thresholds are used with early warning purposes in several countries and regions all over the world, although not in the Philippines so far (Guzzetti et al., 2020).

The use of satellite rainfall data for forecasting landslides is still minimal compared to other rainfall data sources, such as rain gauges or weather radar. Rainfall estimates from satellite products tend to underestimate the rainfall measurements, compared to rain gauge measurements, especially during extreme events (Mazzoglio et al., 2019). This is because rain gauges are nearly point measurements (generally correspond to areas smaller than 1 m$^2$) while satellite measurements are area averaged, for example over an area of 10 x 10 km for the Integrated Multi-satellite Retrievals for the Global Precipitation Measurement

(GPM) Mission (IMERG). Therefore, if a rain gauge is located on the path of a particularly intense convective cell, its records will be significantly higher than measurements from satellite products, which will be averaged over the whole area. Despite these aspects, the usability of satellite products to forecast landslides has been proven, given that the thresholds are derived using the same source of satellite data (Brunetti et al., 2018). In fact, early warning systems based on satellite data are a really powerful tool for developing countries where rain gauges may be scarce or poorly maintained, as well as to implement early

warning systems at regional scales, not just at site-based locations (Kirschbaum and Stanley, 2018; Liao et al., 2010). A clear advantage of the satellite rainfall products is the high spatial and temporal resolution, which enables detailed analysis of rainfall conditions that trigger multiple landslides over large regions.

In some regions of the world, the soil wetness at the beginning of the triggering rainfall has been proven to play a major role

(Bogaard and van Asch, 2002) and therefore to help improve the early warning systems (Guzzetti et al., 2020; Krogli et al., 2018; Marino et al., 2020). Although in previous works in the Baguio area (Philippines) the importance of the antecedent rainfall has been shown to be key to understanding the triggering mechanisms of MORLEs during typhoons (Nolasco-Javier et al., 2015; Nolasco-Javier and Kumar, 2018), no previous work has involved the analysis of soil moisture. The Soil Moisture Active Passive (SMAP) satellite product is a global soil moisture dataset that has potential for analysis of landslide triggering

conditions and early warning (Kirschbaum and Stanley, 2018), but as yet has not been widely used in landslide research.

The municipality of Itogon (Benguet, Luzon) and its surroundings was hit by Typhoon Mangkhut (locally known as Typhoon Ompong) in September 2018, which triggered thousands of landslides, including a fatal one that killed more than 80 composed



of miners and their families (Cawis, 2019). The purpose of this work was: a) to map and characterize landslides triggered by

Typhoon Mangkhut for the first time, producing one of the first complete inventories for a typhoon event in the Philippines, b) to investigate the preconditioning and triggering rainfall and soil moisture conditions and c) examine other factors that made certain slopes susceptible to landslides, d) to consider the potential of satellite based rainfall and soil moisture data for early warning of these regional landslide events.

## 2 Study Area

### 2.1 Geological and geomorphological setting

Our research was conducted over an area of 570 km² at the NW of the Philippines' largest island, Luzon. The study area is located in the province of Benguet (16.19 to 16.31ºN and 120.34 to 120.48ºE), at the Southern end of the Cordillera Central Mountain Range, the largest mountain range of the country (Figure 1). The Eastern half of the study area is characterized by the Upper Agno River course (region 3 in Figure 1), which flows N to S, and is dammed by three cascading dams used for

hydroelectric power generation: Ambuklao Dam in the North, Binga Dam in the middle, and San Roque Dam in the South. In the West, the study area is characterized by smooth slopes between 600 and 1500 m.a.s.l., where Baguio City and the Municipality of La Trinidad are located (region 1 in Figure 1). The mountainous region between Upper Agno River and the city and municipality in the West, in the middle of the study area, was the most affected area by Typhoon Mangkhut in 2018 (region 2 in Figure 1). The valleys are characterized by steep slopes, with altitudes ranging from 263 to 2190 m.a.s.l. in the

highest point. The municipality of Itogon is the main inhabited area in these valleys, with a population of nearly 60000 inhabitants. Itogon is a mining town, where the extraction of gold has been one of the main economic activities since the 1990s, and some tailings dams can be observed in its surroundings. The bedrock of the area is mostly constituted by Cretaceous, Tertiary and Quaternary igneous and sedimentary rocks, part of the magmatic arc formed mainly in response to subduction along the Manila Trench since the early Miocene (Bellon and P. Yumul Jr., 2000). While the sedimentary bedrock consisting

of limestones and clastic sedimentary rocks predominate in region 1 of the study area (Figure 1), the mountainous region in the centre (region 2 in Figure 1) and the eastern river plains (region 3 in Figure 1) mostly consist on diorite and diorite porphyry (MGB, 2006). The study area can also be described as seismically active. The whole area is covered by surficial formations consisting of loam and clays and undifferentiated mountain soils. Finally, the vegetative cover is mainly forest, but it also contains pine trees, fruit trees, shrubs and open grassland (Palangdan, 2018).

### 2.2 Climate

The Philippines is characterized by having several types of climate: from tropical rainforest, tropical savanna or tropical monsoon to humid subtropical, in higher altitudes, such as in our study area. The country is divided in 4 climatic regions, based on the distribution of rainfall as presented in the Modified Corona Climate Map of the Philippines (CADS/IAAS CAD and PAGASA/DOST, 2014). Our study area is in the Type 1 zone, characterized by having two pronounced seasons: dry from





November to April and wet during the rest of the year. Most rain falls between June and September. The average annual precipitation in our study area, during the period 1960-1990 (Hijmans et al., 2005) ranges from 3276 mm in the higher elevations to 1894 mm in the floodplain, with a mean value of 2766 mm. The Western and central part of the study area (Regions 1 and 2 in Figure 1) are characterized by having lower mean temperatures and higher amounts of rainfall. In contrast, in the lower elevations of the river Agno floodplain (Region 3 in Figure 1), it is warmer, and the precipitation rates are lower

(Table 1). The winds are controlled by two systems in the Philippines: the northeast monsoon, active from October to late March, and the southwest monsoon, prevalent during the months of July to September. Both monsoons bring heavy rains in parts of the country where the prevailing wind affects.  Moreover, from the approximately 20 tropical cyclones that enter the Philippine area of Responsibility (PAR) every year, most of them hit northern Luzon, and seven to eight make landfall (Nolasco-Javier and Kumar, 2019; Yumul et al., 2011).

**2.3 Landslide activity related to previous typhoons**

Due to the frequent passage of tropical cyclones over the landslide-prone slopes of the study area, rainfall-induced landslides are frequent. Since 2001, at least 14 typhoons causing landslides have hit the study area, according to Nolasco-Javier and Kumar (2018) and Paringit et al (2020).  The most devastating episodes in the last decades, before Typhoon Mangkhut, have been Typhoon Bilis (2006), with 53 landslides reported; Tyhoon Melor and Typhoon Parma (simultaneous in 2009), with 97

landslides reported and Typhoon Koppu (2015), with 80 landslides reported to the City Disaster Risk Reduction and Management Council (CDRRMC) of the City of Baguio.

During Multiple-Occurrence Regional Landslide Events (MORLEs), such as the ones triggered by typhoons and tropical storms, small or remote landslides are often unreported. Further studies from Nolasco-Javier et al (2015) and Nolasco-Javier

and Kumar (2019) demonstrate that the number of landslides caused by Typhoon Parma in the area of Tublay was, by far, larger than the reported events. Therefore, the actual complete landslide record in the area is unknown. The rapid urbanization, together with the ongoing mining activity also represents a relevant factor in landslide risk in the area, though these factors are were not considered in this study (Mines and Geosciences Bureau, 2018).

**2.4. Typhoon Mangkhut (13-15 September 2018)**

From 13 – 15 September 2018, the study area was hit by the passage of Typhoon Mangkhut (called Typhoon Ompong in the Philippines; Figure 2a). The highest observed 4 day rainfall total (12 to 15 September 2018) of 794 mm in Baguio City PAGASA weather station was recorded due to the passage of Typhoon Mangkhut (Weather Division PAGASA, 2018). The estimations from the Global Precipitation Measurement mission (Huffman et al., 2019) show lower values over the larger area

affected by landslides, with 360 mm of rainfall over a 44-hour period (Figure 2b). The typhoon triggered an elevated number of landslides in the area (Figure 1). The landslides were typically shallow translational landslides, mud and debris flows, often





with a complex behaviour: starting as a shallow landslide and becoming a flow (Varnes, 1978). However, rockslides and rockfalls were also reported. A detailed report on the landslides occurred, followed by a hazard assessment including field surveys in six barangays within critical areas, was issued by the Mines and Geosciences Bureau just after the event (Mines and
Geosciences Bureau, 2018). Some of the debris flows had extraordinarily long runouts, such as the fatal landslide that killed more than 80 miners and their families in the area of Barangay Ucab, on the 15 September around 13:00 h local time, further described in the following sections of this paper.

## 3 Data and methods

### 3.1 Compiling a landslide inventory and magnitude-frequency analysis

The first step to evaluate the preconditioning and triggering factors of the landslides triggered by Typhoon Mangkhut was the creation of a landslide inventory. We mapped landslides manually using satellite imagery by comparing pre- and post- Typhoon Mangkhut images of the study area. The sources of the satellite imagery were of diverse resolution (Table 2) and were combined with digital terrain models as well as with the use of Google Earth™ to more clearly identify the landslides. We experimented with an automatic landslide mapping tool to map landslides more efficiently, however, when comparing with
visual observations, we found the success rate insufficient (Abancó et al., 2020), so the final inventory was entirely done using manual techniques.

Despite having many advantages, such as the possibility to map large and often not accessible mountain regions (Guzzetti et al., 2012), satellite mapping has some limitations, such as the availability of good images, cloud free, within a sensible time
period before and after the event. In our study, high resolution images have a gap of several months (pre- and post- typhoon) (Table 2). Considering that landslides are not uncommon in the area, and that the construction and mining activities are intense, some of the landslides mapped using satellite images may not have occurred during Typhoon Mangkhut but before or after. For this reason, other imagery sources (Table 2) and Google Earth, together with the comparison with local reports reporting field surveys after the Typhoon (Mines and Geosciences Bureau, 2018) have been used to cross-check part of the inventory.

Landslides were mapped as polygons, without distinguishing source and runout areas, as it was often difficult to discriminate between slides, debris flows and earth flows as well as the transition between them. In the cases where the deposition areas were clearly differentiated from failure and runout areas, these have not been included in the polygons; however, in cases where runout was not long it was difficult to differentiate them. The dense vegetation covering a major part of the slopes was
useful to identify and delineate the landslides, as they are easily visible as bare soil within a body of dense vegetation. Moreover, the Normalized Difference Vegetation Index (NDVI) proved to be useful to identify such changes.





We plotted the magnitude-frequency distribution of landslides across the study area and estimated the exponents of the tail of the resulting characteristic power law distribution using the maximum likelihood estimate procedure of Clauset et al. (2009).

Finally, a more detailed analysis was done on one major landslide that occurred in the area of First gate, in Barangay Ucab (region 2 of the study area, see Figure 5), where small scale miners were staying in a Bunkhouse owned by Benguet Corporation (Palangdan, 2018). This landslide had a combined behaviour, evolving from a hillslope into a flow with a particularly long runout, that ended up in a tragedy causing the loss of life of 80 miners.

### 3.2 Analysis of landscape controls on landslides

In order to assess the influence of landscape characteristics on the spatial distribution of landslides, we combined the landslide inventory with topographical data and several thematic maps with terrain information using spatial analysis techniques in ArcMap 10.6.1 (ESRI, 2018). We obtained the frequency distribution for each governing factor both for the total of the study area and only for the areas affected by landslides. A 5-m resolution Digital Surface Model acquired in 2013 with IfSAR techniques (NAMRIA, 2013) provided the topographical information of the study area: elevation, slope and aspect degree.

Maps on soil type (Victor A. Bato, Ozzy Boy Nicopior, 2004), land cover (NAMRIA, 2010) and bedrock geology (MGB, 2006) were used to retrieve information on further terrain factors. Bedrock geology was only available for the 63% of the study area, corresponding to the boundaries of the Baguio quadrangle geological map.

### 3.3 Analysis of rainfall and soil moisture

Rainfall data from 2018 at a resolution of 0.1 degrees (approx.. 10 km) and 30 minute time interval was acquired from the
Global Precipitation Measurement (GPM IMERG) mission (Huffman et al., 2019) for the study area and its surroundings. These data are of particular interest to identify any correlation between the spatial variability of the rainfall associated with Typhoon Mangkhut and the distribution of landslides, instead of having only the point-based data from Baguio city rain gauge (Figure 2). Moreover, it enables tracking of the distribution of antecedent rainfall, and analysis of the role this played in the occurrence and distribution of landslides.


In order to do a more detailed analysis of the rainfall conditions that led to landsliding, the main part of the analysis was based only at one of the rainfall GPM grid points, the nearest to the fatal landslide in Barangay Ucab. The definition of the rainfall duration is a key consideration in the analysis of the rainfall thresholds for landslides, which often brings uncertainty to the analysis (Abancó et al., 2016). Frequently the information of the failure time of landslides is not exactly known, therefore the
duration of the total rainfall (from the start of the rainfall until it finishes) is considered. According to the reports issued after Typhoon Mangkhut, the fatal landslide in Barangay Ucab must have taken place between 05:00 and 07:00 UTC, so in this case, the comparison between the triggering rainfall and the total rainfall was possible.



We analysed the characteristics of the triggering rainfall (Typhoon Mangkhut) as well as other high intensity rainfall events in
the preceding and following months that did not trigger landslides. In order to define rainfall events, we have assumed that an
event starts and ends after and before a period of 1 hour of no rain, following Abancó et al. (2016). Our selection of high
intensity rainfall events was based on a criterion of 3-hour mean GPM IMERG rainfall intensity exceeding 4 mm hr$^{-1}$. The
selection criteria was based on the fact that only 3% of the 30 minute rainfall records from GPM IMERG exceeded the 4 mm
hr$^{-1}$ threshold at the grid point near Barangay Ucab in 2018, therefore the 3-hour time window was considered to be a relevant
filter to select high intensity rainfall events. The purpose of this analysis is to identify the characteristics of the landslide
triggering rainfall by comparing it to other relevant rainfall events and to better identify the conditions that caused Typhoon
Mangkhut to trigger so many landslides.

In addition to calculating antecedent rainfall in the lead up to Typhoon Mangkhut, we also analysed soil moisture data. The
data are also from a satellite source, specifically from the Soil Moisture Active Passive mission (SMAP), acquired by means
of a radiometer (passive) instrument and a synthetic-aperture radar (active) instrument operating with multiple polarizations
in the L-band range. SMAP data have a resolution of 9 km and 3 hours. We used data between May and September 2018, from
Level 4 (L4), corresponding to the surface and root zone soil moisture data (0-100 cm vertical average) in the form of volume
of water/volume of terrain (Reichle et al., 2017).

**4 Results**

**4.1 Landslide characteristics**

A total of 1101 landslides were manually mapped, most of them located in the region 2 of the study area (Figure 1). The
landslides in the study area have areas from 25 m$^2$ up to 120000 m$^2$, representing a mean density of 1.9 landslides/km$^2$, a
maximum value of 4.8 landslides/km$^2$ in region 2.  The fatal landslide in Barangay Ucab, is also located in region 2 and is
highlighted in Figure 1.The exceedance probability distribution of the landslide areas has a characteristic roll-over and  power
law tail. The exponent of the power tail is 2.65, and the rollover point is located at 190 m$^2$ approximately (Figure 3).

Elevations in the study area range between 263 and 2190 m.a.s.l. (Figure 4a) and follow a distribution close to normal, with a
maximum in elevations between 1101 and 1320 m.a.s.l. The landslide density is also highest within the same range, in terms
of mean elevation. However, only 4 landslides occurred below 660 m.a.s.l. and only 1 over 1760 m.a.s.l. Slope gradients that
favoured landslides are shifted towards higher gradients than the study area distribution, with most landslides occurring on
slopes steeper than 30 degrees (Figure 4b). However, flatter areas down to 10 degrees and steeper up to 50 degrees were
affected by failure of landslides too. Particularly striking is the aspect control on landsliding with a concentration of landslides
on East-Southeast-South facing slopes (Figure 4e).






The study area is covered by mountain soils, Ambassador silt loam and Bakakeng Clay mostly. Nevertheless, landslides essentially happen in Bakakeng clay and Halsema clay loam and only to some extent in the Ambassador silt, but not in mountain soils (Figure 4c), which are mostly covered by coniferous forest and natural grasslands. Landslides mostly occurred in wooded grassland, while only a small amount take place in coniferous forests (Figure 4d). It is worth nothing that although the Halsema clay loam is scarce in the study area, the density of landslides is particularly high.

In terms bedrock geology, the area has predominantly a sedimentary sequence of basaltic and andesitic volcanic rocks (Pugo formation), followed by intrusive bodies consisting of diorites and granodiorites (Central Cordillera diorite complex) and a sequence of conglomerates, sandstone and shale (Zigzag formation). The higher density of landslides is located in the Central Cordillera Diorite Complex and the Balatoc Dacite (Figure 4f).

## 4.2 Rainfall and soil moisture conditioning and triggering of landslides

### 4.2.1 Rainfall

The GPM IMERG rainfall data measured in the study area during the passage of Typhoon Mangkhut indicates that the highest intensities, recorded at 03:30 UTC on 15 September, occurred in the eastern region of the study area (Figure 6), which also received the greatest accumulated rainfall over the course of the event (Figure 2). However, the rainfall accumulated throughout the previous two weeks (hereafter called antecedent rainfall) was higher in the central region, where most of the landslides occurred. In this central region, the antecedent rainfall was up to 245 mm (according to GPM IMERG measurements), which is still less than the rainfall accumulated during the Typhoon. The fact that the antecedent rainfall was higher in the area where most of the landslides occurred, even if the intensities were lower, suggests that the wetness of the terrain played an important role in the mechanism of failure.

A detailed analysis of rainfall and soil moisture conditions in the lead up to landslides is based on the GPM IMERG point closest to the fatal landslide at Barangay Ucab (Figure 5) for which timing is most precisely known. The Typhoon Mangkhut rainfall at this point was compared to 33 high intensity rainfall events (3-hour mean intensity above 4 mm hr$^{-1}$) over the preceding and following months that did not trigger a MORLE. While Typhoon Mangkhut rainfall had a duration of 43.5 hours (34 hours until the fatal landslide in Barangay Ucab was triggered), according to the criteria of 1 hour without rainfall for the initiation and end of the event, the durations of the other high intensity rainfall events spanned from 2 to 107 hours. The rainfall that occurred during the passage of Typhoon Mangkhut was also not the highest in terms of accumulated rainfall, as the records show accumulations up to 409 mm in prior high intensity rainfall events. The comparison between the intensity-duration relationships (maximum rainfall intensity for different durations) of the high intensity rainfall events indicates that two events in 2018 had higher intensities (up to 2 hours duration and for long durations of 48 and 72 hours) than Typhoon Mangkhut (Figure 7). Both events happened earlier in the year than Typhoon Mangkhut: on 21 May and 20 July respectively.





As introduced in Section 3.3, the GPM IMERG data represents an average of the rainfall in each of the 0.1 x 0.1 degree cells,
290   which means that even if a high peak of rainfall occurs in a cell (such as the one registered by the rain gauge at Baguio city) it
is averaged above the whole area. We compared the high intensity rainfall events selected for the analysis. The results revealed
that a great number of rainfall events clearly exceed global intensity-duration thresholds (Caine, 1980) and regional thresholds
(Arboleda et al., 1996; Nolasco-Javier et al., 2015; Nolasco-Javier and Kumar, 2018), despite having used data from GPM
IMERG, which tends to underestimate rainfall in extreme events.

### 4.2.2 Soil moisture

The SMAP data containing information on the soil moisture on the root zone (at 0-100 cm depth) from May to September
2018 in 4 different points of the study area was analysed. As can be seen in Figure 8, soil moisture increases from May to
September and the correlation with the rainfall is clear. The increments of soil moisture can be observed in two different
situations: a) after a particular high intensity rainfall, such as the one occurred on 20 July (that had a higher intensity than
300   Typhoon Mangkhut); or b) after periods of more continuous prolonged rainfall at lower intensity, for example in mid-June or
early August. The increase of soil moisture with time is continuous, but especially significant from July onwards. The increase
in July is especially relevant in points C and A, that were lower than B and D in May, June and early July but after this event
are higher. The higher levels of soil moisture achieved in the analysed months are close to 0.455 $m^3/m^3$, which could be close
to the level of full saturation limit of the soil. Typhoon Mangkhut occurred after several days of continuous rainfall, in August
305   and early September, that kept a high continuous level of soil moisture, almost up to 0.45 $m^3/m^3$. Figure 9 shows the timeline
of the rainfall and the soil moisture (in point C) during the typhoon (13 September 2018 at 21:00 UTC until 15 September
2018 15:30 UTC), with a temporal resolution of 30 minutes for the rainfall and 3 hours for the soil moisture.

### 5   Discussion

In this study, we investigate the landscape and meteorological preconditioning and triggering factors of the landslides triggered
by Typhoon Mangkhut in a study area of the province of Benguet (Luzon, Philippines). The typhoon triggered an elevated
number of landslides, with the highest density (4.8 landslides/km$^2$) in the central part of the study area, around the municipality
of Itogon, a region with steep slopes in the southern end of the Cordillera Central. The impact of this event was significantly
high, mainly due to two main aspects: a) an elevated vulnerability of elements in the area, with an important presence of mining
activity and settlements; and b) the complex behaviour of some of the landslides, with long runouts and elevated entrainment
rates, which magnified their volume. We will look at landslide runout and controls on this in a separate study.

Using a landslide inventory based on a single event provides information that is strongly influenced by the event characteristics
itself but may not be representative for the evaluation of the landslide hazard of the area. Systematic inventories should be





conducted over multiple years to provide more reliable information for the evaluation of size statistics of landslides as well as

of the susceptibility of the landscape to landsliding (e.g.: Guzzetti et al., 2005, 2006; Del Ventisette et al., 2014). However, our analysis gives an indication of landslide characteristics and of the landscape controls in the region that will contribute towards a future landslide hazard assessment. Furthermore, it is of great importance for the understanding of the rainfall triggering conditions of landslides in the territory and in working towards landslide early warning in the region.

**5.1. Landslide characteristics and landscape preconditioning**

We are not aware of any magnitude-frequency distribution of landslides for the Philippines and thus we fill this gap in the literature here. The magnitude-frequency distribution of the areas of the landslides in the inventory shows a characteristic shape with rollover and power law tail, with an exponent of the power tail of 2.65 and a rollover point around 190 $m^2$. This exponent is higher than two landslide distributions triggered by typhoon events in Taiwan, with exponents of 1.42-1.60 (Chien-

Yuan et al., 2006), though similar to earthquake-triggered landslide inventories in China (Li et al., 2013) and Haiti (Gorum et al., 2013): 2.63 and 2.71 respectively (Tanyaş et al., 2019). These numbers, as 2.65 obtained in this study, suggest that the small landslides are more frequent than larger ones, in comparison to other studies where the exponent of the power law tail is lower than 2 (Bennett et al., 2012; Van Den Eeckhaut et al., 2007). Further mapping in the region and across other regions of the Philippines will help to refine these distributions and exponents.


An interesting finding of this study is the strong aspect control on landsliding in Typhoon Mangkhut. A possible explanation for aspect control of landslides in the literature is differences in vegetation and thus root cohesion between aspects that receive differing amounts of solar insolation (Rengers et al., 2016). Indeed landslides in the region occur within wooded grasslands and rarely in coniferous forest (Figure 4d), with higher root binding of the soils. Another possible explanation would be that

these slopes have a soil type that is more prone to landslides, so the Bakakeng clay or the Halsema clay loam. This reddish-brown clay and the brownish clay loam are characterized by having a very slow internal drainage (low permeability) (Carating et al., 2014), which may explain their tendency to fail, due to an excessive pore pressure, when they are saturated. Further analysis on the geotechnical properties of these clayey soils should be carried out to determine what makes them more prone to landsliding to the other soils in the study area. For example, in high plasticity clays, their exposure to repetitive wet-dry

cycles may reduce their shear strength (Khan et al., 2017); or the appearance of cracks, which may change the hydraulic conductivity and make them more prone to landsliding (Khan et al., 2019).

However, although landslides tended to happen in a certain land cover (wooded grasslands) and soil type (Bakakeng clay and Halsema clay loam), there is no evidence that this could explain the prevalent orientation of the slopes affected by landslides

to the East-Southeast, as there are no distinct differences in vegetation or soil type between different aspects. Therefore, we suggest this trend may be better explained by the typhoon trajectory and wind directions, from East-Southeast to West-Northwest, with winds driving rains into windward facing slopes more intensely, whilst leeward slopes are more protected.





On one hand, in some previous studies typhoon trajectory and wind direction have not been found to play a role in typhoon triggered landslides in similar regions (e.g. Chen et al., 2019). On the other hand, other studies suggest that the wind does
affect rainfall intensities on various slope aspects as the leeward sides are subjected to lower rainfall intensity than the windward sides, and therefore the occurrence of landslides can be affected by the winds (de Lima, 1990; Liu and Shih, 2013). Hence, the topic deserves further research.

Although in this study we have not considered anthropogenic factors, local reports (Mines and Geosciences Bureau, 2018) and
studies (Nolasco-Javier and Kumar, 2018) have pointed out that rapid urbanization and mining activities can severely impact the susceptibility of the slopes to landsliding. The presence of underground mines in the region, summed to the existence of faults and fractures in the bedrock, generates a labyrinth of underground excavations that may clearly affect the slope stability and could be looked at in further research.

**5.2. Rainfall and soil moisture conditions leading to landsliding**
The analysis of 2018 rainfall in the study region shows that more rainfall intensity does not mean more landslides, in contrast to some other studies (Chen et al., 2013; Lin and Chen, 2012). In fact, the results of this study do not support the model proposed by Crozier (2017), that suggests a higher density of landslides at the core of the rainfall intensity cell, decreasing as the rainfall intensity does. Instead, the density of landslides was higher in areas of greater antecedent rainfall, supporting the
findings of Nolasco-Javier and Kumar (2018) in the same region. However, whilst Nolasco-Javier and Kumar (2018) suggested a threshold of 500 mm of rainfall accumulated over the rainy season is needed for landsliding, we suggest that the threshold is far higher. The 500 mm threshold was already reached in our study area on the 14 June 2018, yet a very intense event in July and some other intense rainfalls failed to trigger landsides according to the records and the satellite imagery available. Our study suggests a threshold of 2600 mm of rainfall accumulated over the rainy season for landsliding to occur. This demonstrates
the difficulty of defining thresholds based on rainfall for a given area and the need for multiple events to refine thresholds. Futhermore, multiple rainfall events exceeded the global threshold Caine (1980). This is probably because such thresholds are obtained using a high diversity of meteorological patterns, therefore may be too low for extreme climates such as tropical in the Philippines.

Soil moisture data provides an even more accurate picture of the soil conditions at the time of landsliding than antecedent rainfall data. There are several studies that have started to combine soil moisture with rainfall data to define landslide thresholds (Hürlimann et al., 2019; Mirus et al., 2018) and for landslide early warning (e.g.: Kirschbaum and Stanley, 2018; Krogli et al., 2018). The analysis of soil moisture in our study area in the lead up to Typhoon Mangkhut shows that the volumetric water content of the soil increased over the rainy season, reaching a maximum of 0.455 $m^3/m^3$, when the typhoon happened in
September 2018. This value is actually a reasonable value for the porosity of clays, which would suggest that it may correspond





to the saturation point of Bakakeng clay. Therefore, any rainfall occurring in these saturated conditions would create an increase of soil pore-water pressure, which would result in a decrease of effective stress and therefore a tendency to fail.

### 5.3. Potential of satellite-based rainfall and soil moisture data for landslide early warning

In order to explore the potential of satellite-based rainfall and soil moisture data for landslide early warning, we conducted a further analysis of rainfall and soil moisture conditions for the 34 high intensity rainfalls (incuding Typhoon Mangkhut) (Figure 10). Although Typhoon Mangkhut rainfall shows one of the highest values of volumetric water content in the soil and also has high values for all four rainfall parameters analysed (mean rainfall, peak rainfall intensity, rainfall duration and total rainfall of the event), it does not clearly stand out from other rainfalls. This may be because there are other factors involved in

landsliding that we have not considered here, such as the atmospheric pressure (Pelascini et al., 2020) or the wind directions, linked to the Typhoon trajectory. Alternatively, it may be that satellite-based rainfall and soil moisture data do not adequately capture the conditions on the ground. Hence, satellite-based data should be used with caution in landslide early warning systems (Hidayat et al., 2019; Kirschbaum and Stanley, 2018), ensuring that threshold curves are derived using the same source of data (Brunetti et al., 2018).


Further work should be carried out in the region in order to establish a reliable threshold to identify and provide reliable early warning of landslide-triggering rainfalls, using either rain gauges or satellite rainfall products or a combination. The high temporal resolution of satellite data allows more detailed thresholds, which would be more useful to be applied in early warning systems, than daily values, such as the ones suggested by Nolasco-Javier and Kumar (2018) and Nolasco-Javier et al. (2015).

However,it is also important to consider the uncertainty that the satellite data brings compared to in-situ measurements. Hence, we are working on the installation of in-situ sensors to verify satellite data. A combination of satellite rainfall and soil moisture data (in real time or forecasted) with rain gauges and soil-moisture sensors could potentially be combined in a future landslide early warning system.

## 6    Conclusions

We used satellite imagery to produce a complete inventory of landslides triggered by Typhoon Mangkhut (2018), which contains 1101 landslides. The magnitude-frequency distribution of the landslide areas, the first we are aware of for the Philippines, has a characteristic rollover effect, with a power law tail, with an exponent of 2.65. The exponent is higher than in other typhoon-triggered landslide inventories, which suggests that bigger landslides are rarer in the study area.

The geomorphological analysis of the inventory shows that most of the landslides happen face East-South East. After discarding land cover or soil type related explanations, we suggest this is somehow controlled by the trajectory of the typhoon and wind directions, from East-Southeast to West-Northwest. Landslides occurred predominantly in Bakakeng clay and

Halsema clay loam, two clayey soils that cover some slopes in the study area, which have a low permeability. Another factor that may have played a role in the distribution of landslides but was not considered in this research is extensive mining activities

in the region, associated to the excavation of many interconnected underground mines over the study area, and the increasing infrastructure, which generates overloads in the slopes.

We used GPM IMERG rainfall data to analyse the spatial distribution of the rainfall associated to Typhoon Mangkhut. Antecedent rainfall in the two weeks leading up to the typhoon better explains the spatial landslide pattern than rainfall

intensity. This result suggests, as pointed out by other studies, that the soil moisture may play a very important role in the trigger of landslides in the area. We used SMAP-L4 soil moisture data to analyse the soil moisture evolution along the rainy season of 2018. The results show that soil moisture increased along the season, achieving highest values (probably at the saturation point) when Typhoon Mangkhut hit the area. However, in previous months, other intense rainfalls happened, also with high volumetric water content, that did not trigger landslides.


The Typhoon Mangkhut rainfall was compared to 33 other high intensity rainfalls occurred in 2018 and to some published global and regional rainfall thresholds. Firstly, of this analysis was that, although satellite-based rainfall products tend to underestimate rainfall measurements, a great number of rainfall events (that did not trigger landslides) were above global and regional rainfall thresholds used for comparison. The need of further analysis of landslide triggering rainfall in the area is

highlighted, preferably including a comparison with ground-based measurements.

Finally, we did a preliminary analysis to foresee the potential of combining triggering rainfall and the soil moisture data to be used in a potential early warning system. We find that it is difficult to isolate Typhoon Mangkut from other rainfall events that happened in the lead up to the typhoon with higher intensities and under equally saturated soil conditions, yet did not trigger

landslides. The results show that it is not possible to draw a threshold only using one single landslide triggering event and point out that the exclusive use of satellite data may induce some uncertainties due to the area-averaged measurements, which need to be analysed in future studies.

**Code/Data availability**

The landslide inventory will be available at the end of the SCaRP project (currently January 2022), and will be accessible at NERC Environmental Information Data Centre EIDC  https://eidc.ac.uk/deposit. In the meantime, anyone interested in a collaboration using the data may contact the corresponding author.



**Author contribution**

GB and CA designed the study. CA mapped the landslides, analysed the data (geomorphological, rainfall and soil moisture)
and led the writing of the article with contributions from GB. AM provided GPM IMERG rainfall data and contributed in the
analysis of this. MM and FT provided information on the Mangkhut event and contributed in the collection of
geomorphological data. All authors contributed in the writing and gave final approval of this manuscript.

**Competing interests**

The authors declare that no competing interests are present.

**Acknowledgments**

We are thankful to Goddard Space Flight Center, Precipitation Measurement Missions at the National Aeronautics and Space
Administration (NASA) and the National Snow and Ice Data Centre respectively for making GPM IMERG and SMAP datasets
freely available. David Hein-Griggs is thanked for his support in the extraction and processing of SMAP-L4 data.
We are also grateful to the Mines and Geosciences Bureau (MGB) for providing the Geological Map from Baguio region, the
Philippine Atmospheric, Geophysical and Astronomical Services Administration (PAGASA) for providing the rainfall data of
the Baguio rain gauge and to Xavier Fuentes for making the soil and land cover data available through PhilGIS.org. The
researchers based in the UK (C. Abancó, G. Bennett and A.J. Matthews) are funded by NERC Newton Agham fund (contract
NE/S003371/1 Project SCaRP) and the researchers based in the Philippines (M. Matera and F. Tan) are funded by the
Department of Science and Technology – Philippine Council for Industry, Energy, and Emerging Technology Research and
Development (DOST-PCIEERD, Project No. 07166).



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



**Tables**

**Table 1: Average maximum and minimum monthly precipitation (from 1960 to 1990) in the different regions of the study area (see Fig.1). Source: https://www.worldclim.org/, version 1.4, release 3.**

|  | Region 1 | | Region 2 | | Region 3 | |
|---|---|---|---|---|---|---|
| Month | Average max precipitation (mm) | Average min precipitation (mm) | Average max precipitation (mm) | Average min precipitation (mm) | Average max precipitation (mm) | Average min precipitation (mm) |
| January | 22 | 3 | **30** | 3 | 18 | 3 |
| February | 19 | 2 | **22** | 2 | 17 | 2 |
| March | **50** | 30 | 46 | 23 | 45 | 29 |
| April | **115** | 68 | 101 | 63 | 109 | 69 |
| May | 247 | 190 | **265** | 219 | 256 | 203 |
| June | 385 | 249 | **432** | 337 | 417 | 310 |
| July | 610 | 346 | 617 | 331 | **635** | 379 |
| August | 685 | 415 | **783** | 546 | 739 | 506 |
| September | 487 | 307 | **586** | 388 | 556 | 351 |
| October | 316 | 165 | **335** | 197 | 329 | 217 |
| November | 174 | 65 | **192** | 55 | 170 | 65 |
| December | 58 | 20 | **78** | 13 | 55 | 20 |

**Table 2: Details of the satellite imagery sources used in this study.**

| Imagery source | Spatial resolution (m) | Data image pre-typhoon | Date image post-typhoon |
|---|---|---|---|
| WorldView2 | 0.5 | 18/02/2018 | 02/03/2019 |
| Sentinel 2 | 10 | 28/04/2018 | 09/11/2018 |
| Planet Labs | 3 | 01/08/2018 | 21/09/2018 |






**Figures**

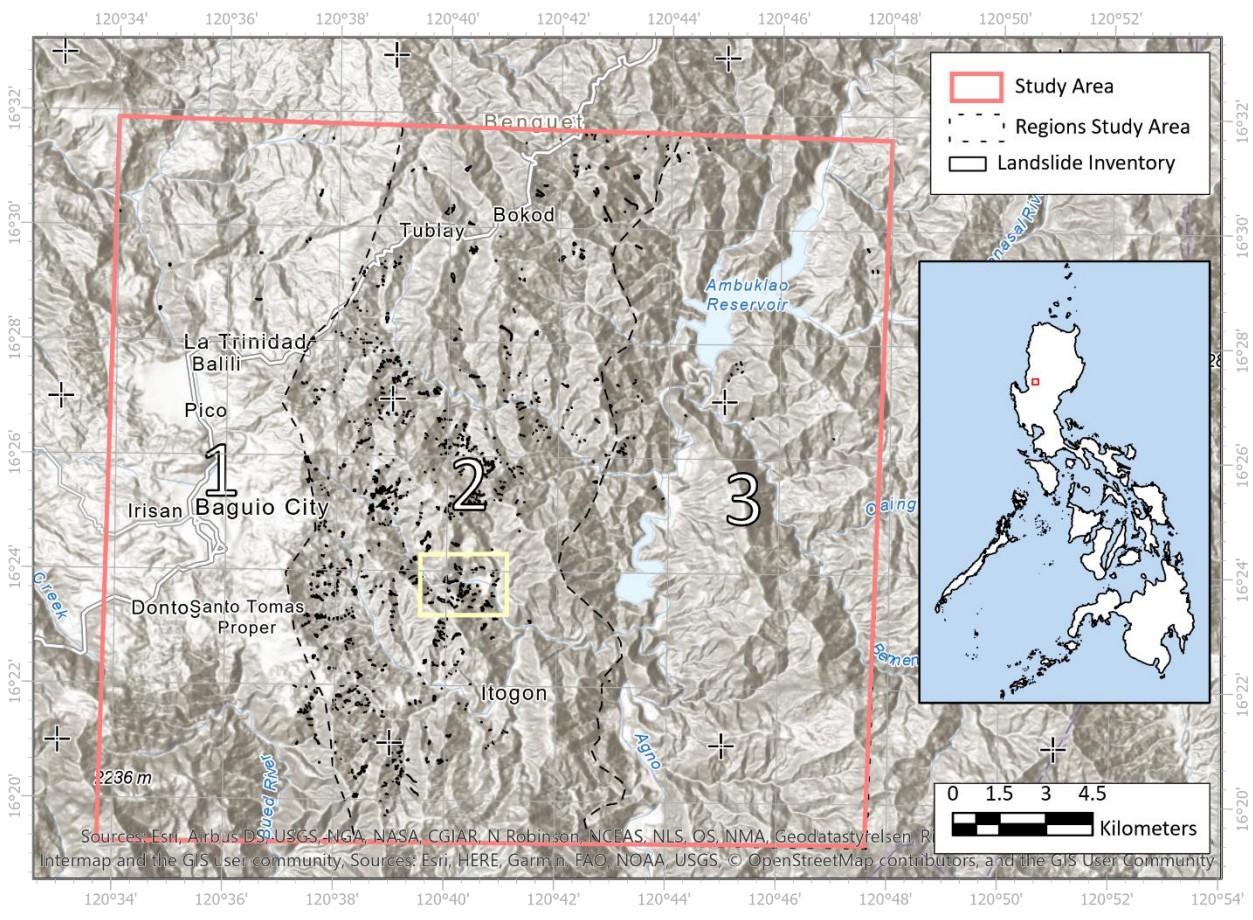

**Figure 1: Landslide inventory. Area affected by 1096 landslides triggered by Typhoon Mangkhut was mapped using polygons. Three regions of the study area are distinguished (see text). The yellow area is referred in Figure 5. Inset, location of the study area within the Philippines, in the province of Benguet (Luzon).**



a)

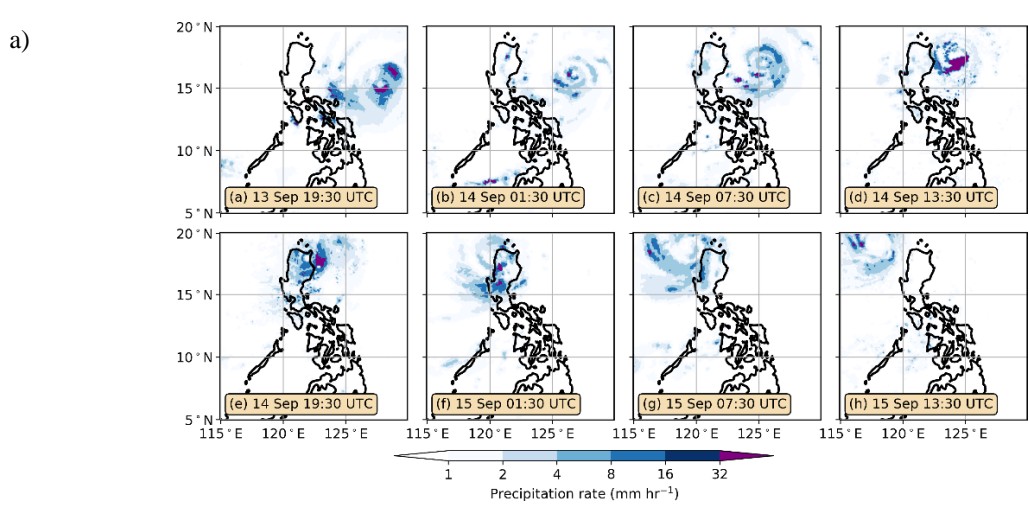

b)

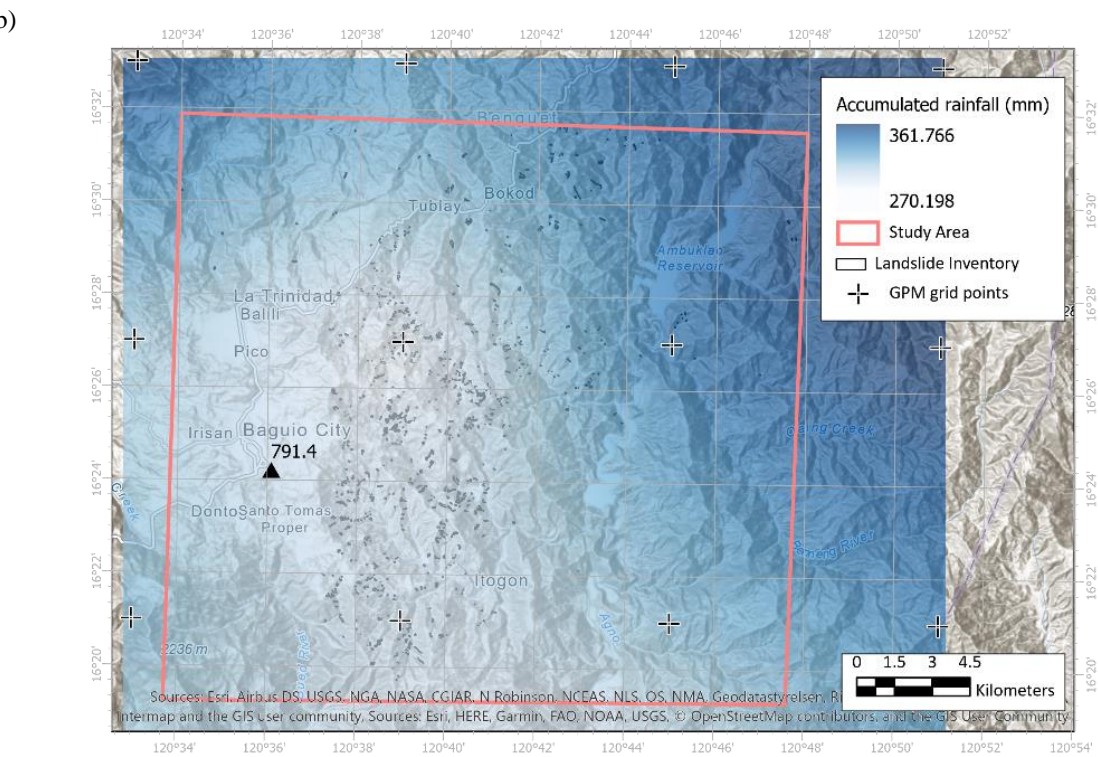

**Figure 2: a) GPM IMERG data showing the evolution of Typhoon Mangkhut over the Philippines on the between 13 and 15 September 2018; b) Accumulated rainfall during Typhoon Mangkhut (13/09/2018 19:30 to 15/09/2018 15:30 UTC) within the study area and its surroundings, according GPM IMERG data (map) and rain gauge records in Baguio city (black triangle).**

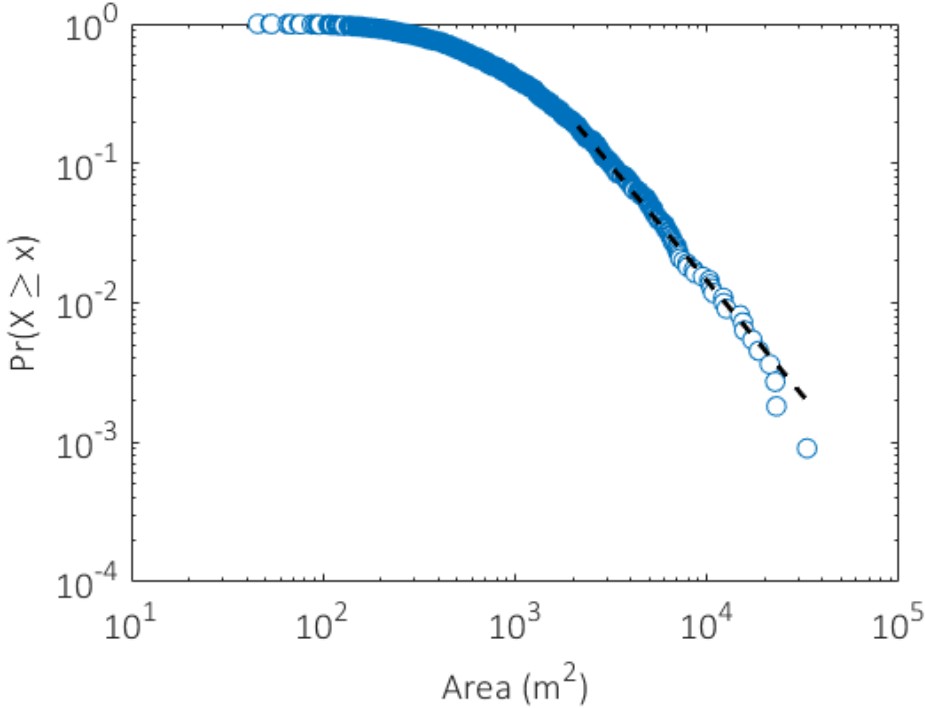


**Figure 3: Exceedence probability distribution for the 1101 landslide areas in the inventory, fit with theoretical power law model by the maximum likelihood method.**




**Figure 4: Histograms of different geomorphological parameters over the study area and frequency of landslides for every parameter class. The geomorphological parameters are: a) Elevation, b) Slope; c) Soil type; d) Land cover type; e) Aspect; f) Geology. Note that Geology is only over the 63% of the study area and 93% of the landslides as the Geology Map was only available for part of the area.**





a)

b)

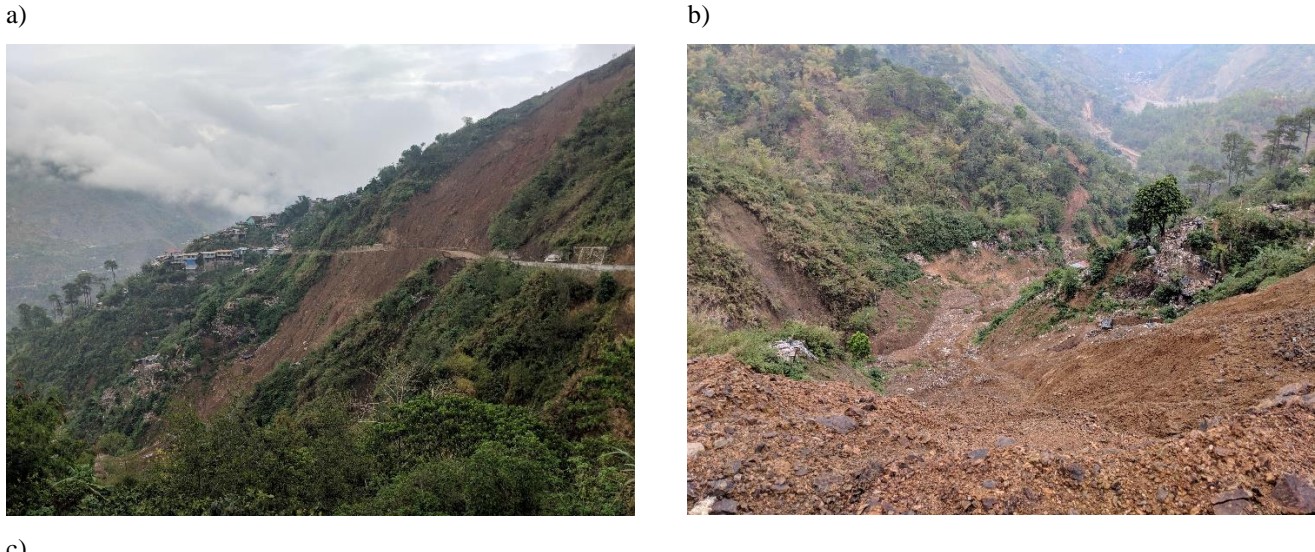

c)

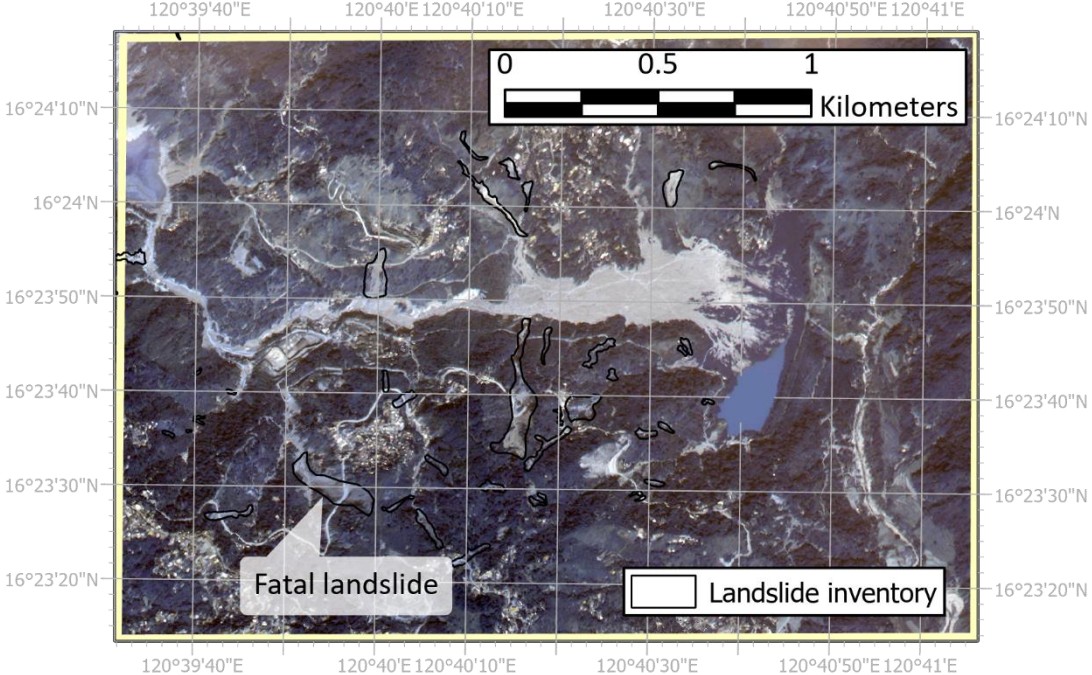

**Figure 5: a) General view of the initiation area of the fatal landslide in Barangay Ucab that killed more than 80 miners; b) view of the particularly long runout of the landslide, from the road facing downhill and c) location of the landslide (see yellow rectangle in Figure 1 for the exact location within the study area). Satellite imagery from Worldview (02/03/2019).**


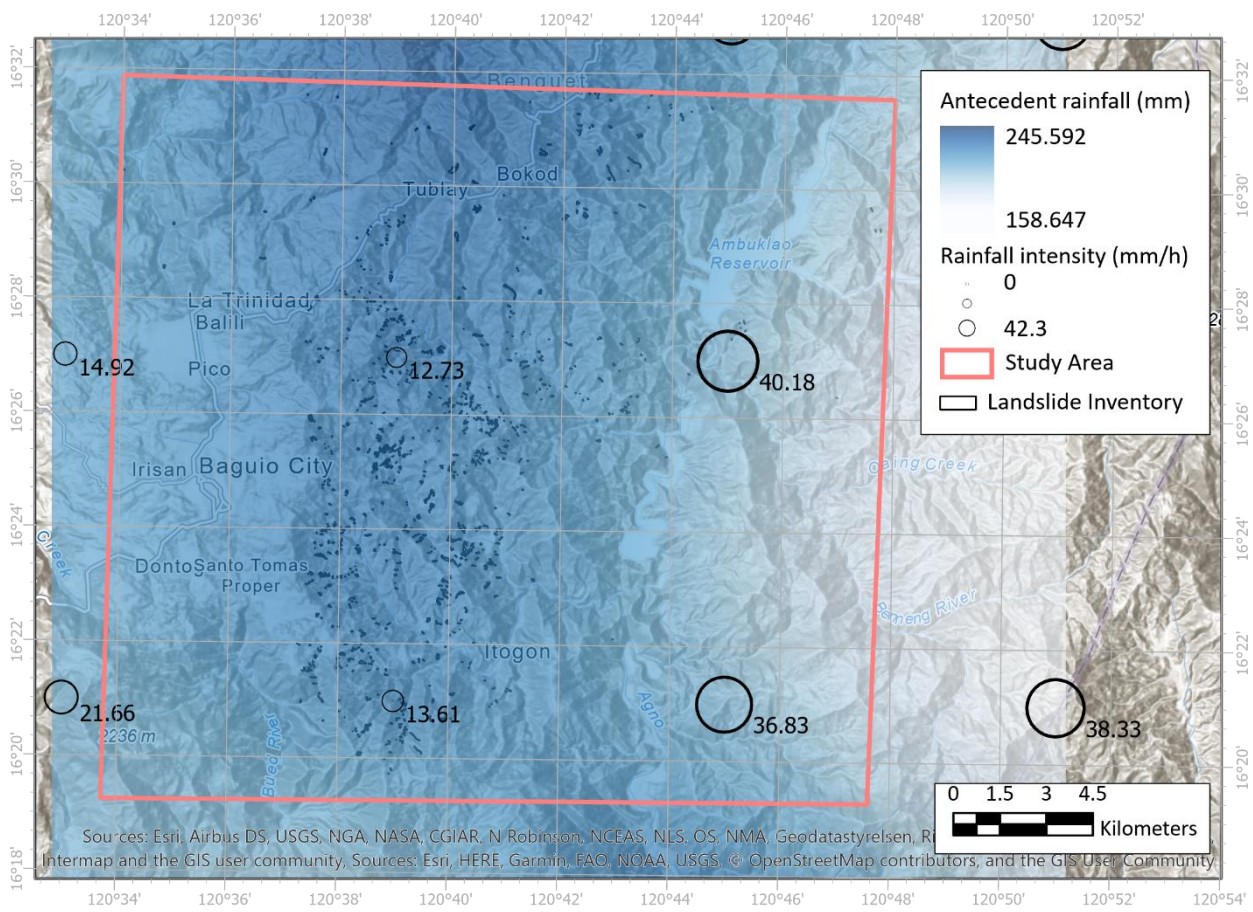

Figure 6: Colour gradient represents the antecedent rainfall, accumulated during the 13 days before the Typhoon. The values in the GPM IMERG grid points indicate the rainfall intensity on the 15 September 2018 at 03:30 UTC (maximum intensity in the study area)




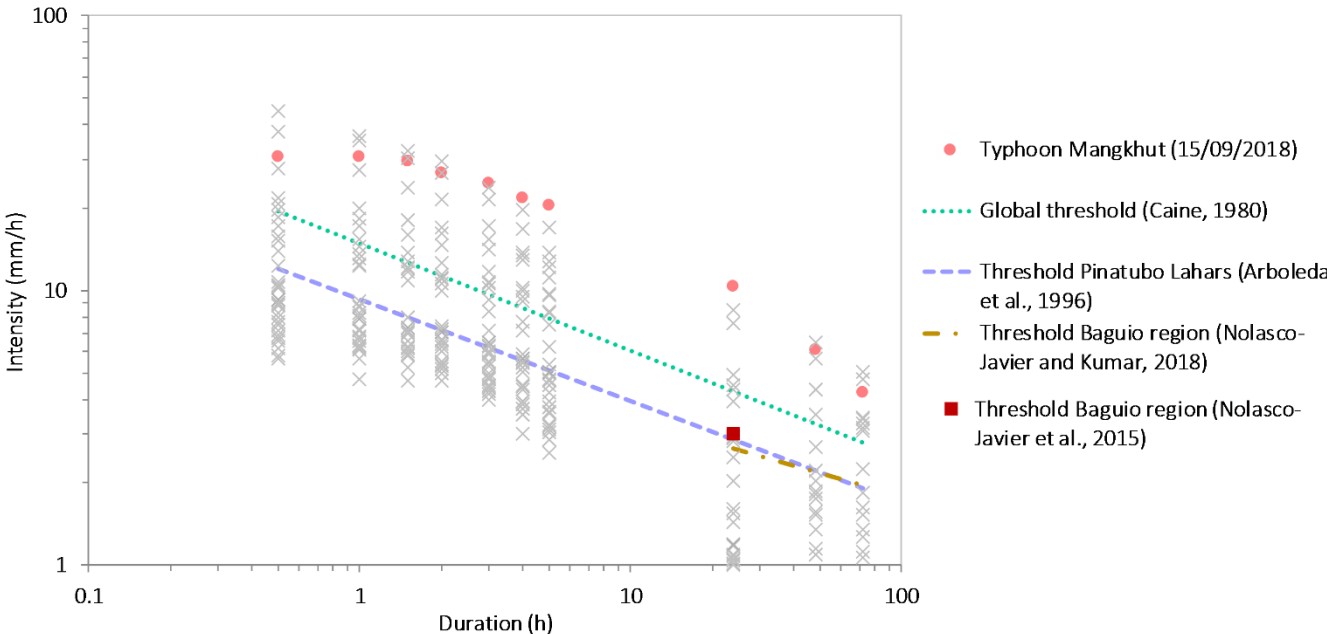


**Figure 7: Intensity-Duration correlations of rainfall associated to Typhoon Mangkhut as well as 33 high intensity rainfalls along 2018 in the study area. The rainfalls are compared to some global and regional thresholds published in the literature.**



a)

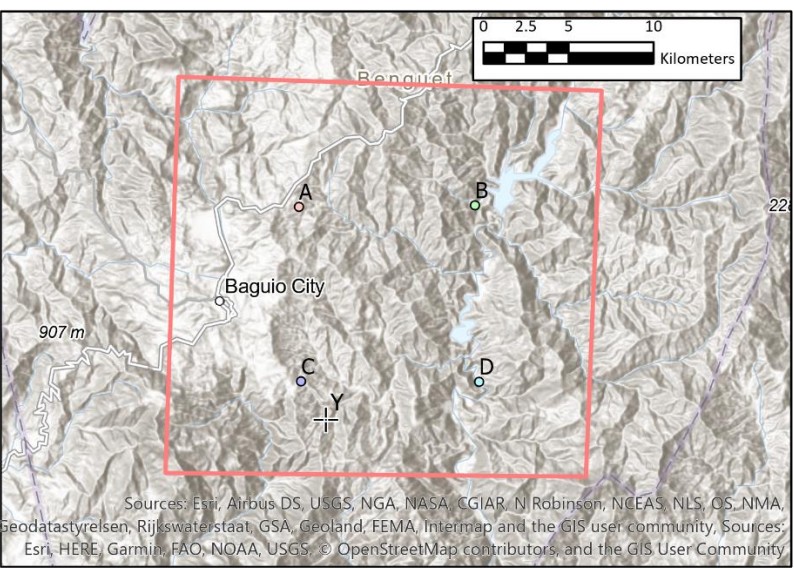

b)

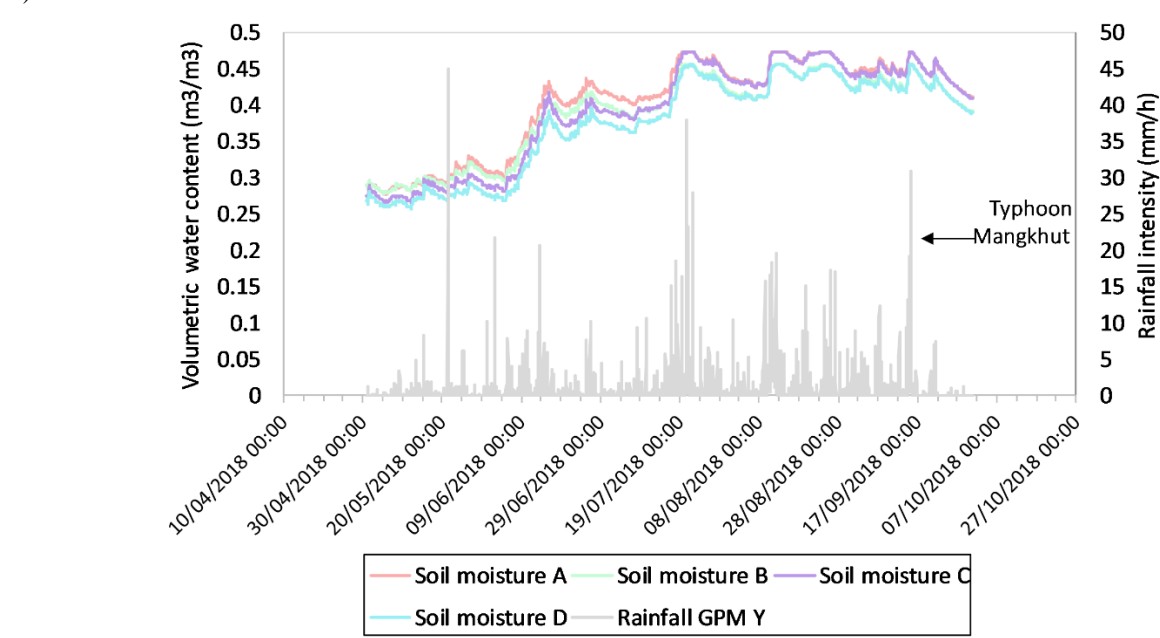


**Figure 8: a) Location of SMAP-L4 and GPM IMERG grid points from which the soil moisture and rainfall data has been obtained within the study area and b) evolution of soil moisture (in volumetric water content) in A to D and rainfall in Y from the beginning of May to the end of September 2018.**



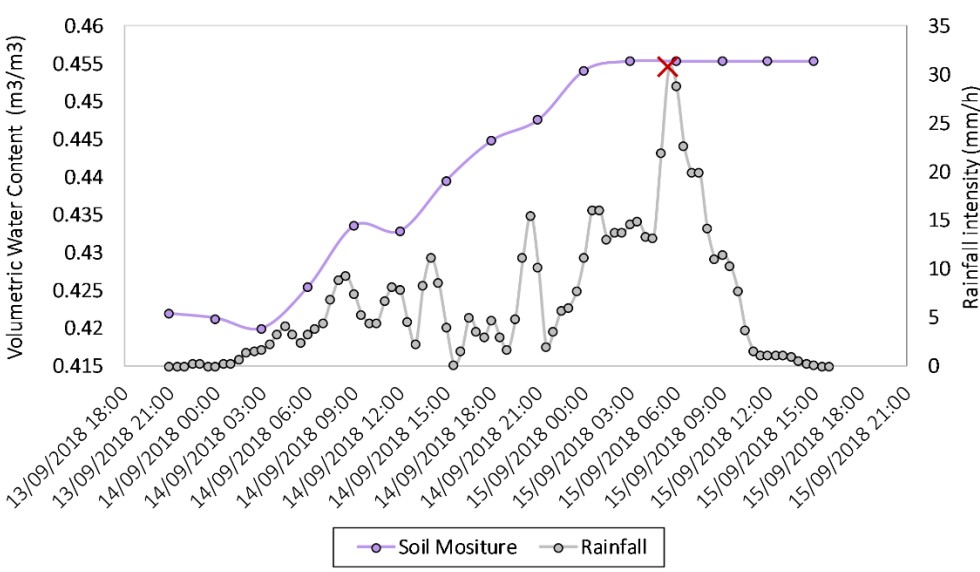


**Figure 9: Detail of rainfall (GPM IMERG) and soil moisture (SMAP-L4) in points Y and C (see Figure 8a) respectively, during Typhoon Mangkhut. The red cross indicates the estimated time of the landslide occurrence (in UTC).**








**Figure 10: Correlation between soil moisture a) mean rainfall intensity of the event (total rainfall/rainfall duration), b) peak rainfall intensity during event, c) duration of the rainfall and d) acummulated rainfall during the event for the 33 (and Typhoon Mangkhut) high intensity rainfalls in the study area (see Figure 7). Soil moisture data obtained from SMAP-L4 data at point C and rainfall data from GPM IMERG at point Y (see Figure 8).**

