# Peer review of "The role of geomorphology, rainfall and soil moisture in the occurrence of landslides triggered by 2018 Typhoon Mangkhut in the Philippines"

_Natural Hazards and Earth System Sciences, 2020_

## Referee Comment (RC1) · Anonymous Referee #1 · 18 Sep 2020

The study concentrates on predisposing and triggering factors that caused a large population of landslides during the 2018 Typhoon Mangkhut, in the Philippines.

The manuscript is quite clear and not difficult to read. The structure can be a bit improved, but already acceptable. Data and analyses are of some interest but there are weaknesses, in particular in the critical evaluation of the result, that makes the discussion in some parts superficial and ambiguous, also the literature review in the introduction can be improved with some more references better linked to the topic.

My main concerns are:

I have not found any clear connection between the meteorological information and the geo-environmental information as if they were two different topics. Is that a choice because lack of connection evidence? Are the resolutions too different for a proper comparison?

Some points related to the sampling methods are unclear (to me), need for further clarifications

Some of the comparisons between the results obtained in this work and elsewhere should be better contextualized, in particular when obtained with different methods or data.

Connected to the previous point, results are depending on some of the preliminary assumptions, including the definition of rain event. This is ok, but the impact of the choice should be better discussed, and can we compare products obtained with different definitions of the initial settings?

Specific comments

Introduction

25 46% of rainfall-triggered landslides: I suggest to add 'known', probably that database is incomplete

30 in the Philippines has still received little attention: do we know why? Just a curiosity

30 results in hazard and risk assessment techniques: what techniques are the Authors referring to?

35 landslide inventories are often not available due to incomplete event records: not sure I understand. Are the inventory missing, or incomplete?

40 Actually the relative literature is very rich, and I don't think that this list is adequate

to intercept the concept.

40 It is particularly challenging in regions hit by the passage of typhoons...: why more challenging?

45 landslide magnitude distribution: is it true for all the types of landslides?

65 geographical distribution: sorry, of what? Do the authors mean the size of the study area?

70 resolution: uncertainty on the resolution. Can the Authors please unravel this concept?

80: about this concept, I suggest you have a look at here.

85 high spatial: compared to? I actually suggest to clarify the concept, since it was told that the resolution is not enough to catch convective processes. Is the fact of working over large areas enough to consider the problem not impacting on the final results?

85 In some regions of the world: it is too generic, isn't it?

100 for the first time: do move it earlier

100 producing one of the first complete inventories: this sentence sounds a bit weird to me: how can more than one complete inventory exist? How do the Authors define 'complete' (and how verified...)

2 Study Area 2.1 Geological and geomorphological settings

110 steep slopes: I suggest to provide here some quantitative info about the slope

120 The study area can also be described as seismically active: according to....

2.2 Climate

2.3 Landslide activity: not sure I would use 'activity' here

140 simultaneous in 2009: overlapping or in two different parts?

150 The rapid urbanization, together with the ongoing mining activity also represents a relevant factor in landslide risk in the area, though these factors were not considered in this study (Mines and Geosciences Bureau, 2018). : this sentence, here, is out of context.

2.4. Typhoon Mangkhut (13-15 September 2018)

155: The highest observed 4 day rainfall total (12 to 15 September 2018) of 794 mm in Baguio City PAGASA weather station was recorded due to the passage of Typhoon Mangkhut (Weather Division PAGASA, 2018).: I suggest to re-phrase

3 Data and methods

3.1 Compiling a landslide inventory and magnitude-frequency analysis

170 We experimented with an automatic landslide mapping tool to map landslides more efficiently: this sentence is too generic and probably in the wrong place (maybe discussion?). What tool? Set up? How did the Author manage the different images? What does the success rate was low mean? And compared to what?

175 limitations: or problems? I also suggest to refer to optical remote sensing here, SAR is starting to provide some alternatives, also for such large events. I suggest to have a look at:

Measures of spatial autocorrelation changes in multitemporal SAR images for event landslides detection (2017) AC Mondini. Remote Sensing 9 (6), 554.

180 have been used to cross-check part of the inventory: and the result is...

3.2 Analysis of landscape controls on landslides

200 governing: I suggest to change the term

3.3 Analysis of rainfall and soil moisture

215 total rainfall (from the start of the rainfall until it finishes): It sounds contradictory

with "These data are of particular interest to identify any correlation between the spatial variability of the rainfall associated with Typhoon Mangkhut and the distribution of landslides, instead of having only the point-based data from Baguio city rain gauge". I suggest to better specify that you are doing a zoom in.

220 total rainfall (from the start of the rainfall until it finishes): I suggest to define here what start and end mean

225 did not trigger landslides: this is a critical point, how to make sure that landslides did not occur? Please, do see a previous comment on the inventory check.

225-230 relevant: the previous numbers are objective, this sentence is relevant. I suggest to explain better why this is relevant. Furthermore, can this rule be applied to all the other events? Perhaps rules should change according to the type of rain event... 4 Results 4.1 Landslide characteristics

General questions:

1) for every single landslide, all the pixels were used, or only one 'representative'? 2) are the pixels inside and outside landslides comparable in number? Normalized?

245 figure 3: actually difficult to see the rollover. I recommend to add information about the quality of the fit (uncertainty)

245 maximum in elevations: the peak of the normal distribution?

4.2 Rainfall and soil moisture conditioning and triggering of landslides 4.2.1 Rainfall

275 timing: time of occurrence?

4.2.2 Soil moisture

5 Discussion

315 Using a landslide inventory based on a single event provides information that is strongly influenced by the event characteristics itself: I suggest to re-phrase.

**5.1. Landslide characteristics and landscape preconditioning**

325 we fill this gap in the literature here: it is actually an example, I would be a bit less 'absolute'. . ..

330 small landslides are more frequent than larger ones: in fact, it is difficult to see the rollover. I think this comparison is interesting, but it is missing a few of elements: type of used data, mapping methods, fitting procedures (the constrains imposed by the fitting), and others should be better unraveled.

330 Further mapping in the region and across other regions of the Philippines will help to refine these distributions and exponents: this sentence sounds a bit weird. The distribution here is related to this event and in the local geo-settings, while the sentence seems to look for a general behavior.

335 aspect: how about anaclinal, cataclinal??

360: how local are these effects?

**5.2. Rainfall and soil moisture conditions leading to landsliding**

365-380: I have some concerns about this sub-paragraph because it is unclear to me whether the different results can be really compared since the definition of the events are different, but more critical, the data are eventually different!!

385 This value is actually a reasonable value for the porosity of clays,: perhaps, but I think this should be supported by evidence, papers, references..

**5.3. Potential of satellite-based rainfall and soil moisture data for landslide early warning**

395 Alternatively, it may be that satellite-based rainfall and soil moisture data do not adequately: see one of my previous comments. Not adequate, or not comparable...

2020-259, 2020.

---

## Referee Comment (RC2) · Xiangzhou Xu (Referee) · 9 Oct 2020

Landslide plays an important role in landscape evolution, delivers huge amounts of sediment to rivers and seriously affects the structure and function of ecosystems and society. This paper, which is entitled "The role of geomorphology, rainfall and soil moisture in the occurrence of landslides triggered by 2018 Typhoon Mangkhut in the Philippines", tries to examine the factors susceptible to landslides, consider the potential for early warning of the landslides. The topic looks very interesting and valuable. Nevertheless, a major revision is needed before the manuscript is accepted for publication in

the journal NHESS.

Some problems are listed as follows: 1. Title: I suggest you erase the words "triggered by 2018 Typhoon Mangkhut" in the title. I think the readers will be interested in a relatively universal law related to landslides instead of only a certain storm. Also you have to add an in-depth discussion corresponding to the title revision. 2. Abstract: This part has to be rewritten. The point, "a) it was one of the most intense rainfall 20 events in the year but not the highest", is a condition to induce the landslides instead of a result. In addition, have you resolved the problems presented in lines 99-104 (page 4), Section Introduction? Please let me know in the abstract with a concise description. 3. Study area. Too many details have been given in Section 2. You may delete some descriptions which are not closely related to the topic of the paper, and the subtitles of the section, including subtitles 2.1-2.4. 4. Methods. (1) Line 110, page 4: What's the meaning of the unit "m.a.s.l"? (2) How to distinguish a landslide in the area with scarce plants? (3) I do not think a rainfall with the intensity of 4 mm hr-1 is intensive rainfall. Line 374, page 12. You said "Our study suggests a threshold of 2600 mm of rainfall accumulated over the rainy season for landsliding to occur". However, in lines 226-227, "Our selection of high intensity rainfall events was. . . exceeding 4 mm hr-1". The intensity of 4 mm hr-1 is really too small. Maybe 4 mm min-1? 5. Discussion. Section 5.1. What's the meaning of preconditioning, and have you discussed the preconditioning here? 6. Figures and tables. (1) Most of the figures are not clear. The sizes of the texts in some figures are too small. The figures should be clear as they are printed in black and white. (2) Figure 1. What are the differences between the "study area" and "Regions Study Area"? (3) Figures 2 and 8. The general titles of the figures are needed. (4) Figures 9 and 10. The scales of the vertical coordinates are anticipated. (5) The format of the table in the manuscript is not suitable to the requirements of the journal NHESS. 7. The English writing of this paper is readable. Nevertheless, still some minor language errors exist, e.g., the word "are" in line 153 of page 5 should be erased; in line 51 of page 2, the words ", however the. . ." may be replaced with the words "; however, the. . ."; the first letter of the word "Clay" in line 256 of page 9, may

be in lower case.

---

## Author Comment (AC1) · 17 Dec 2020

**Response to anonymous reviewer (RC1)**

We are grateful for the very detailed reading of the paper and substantive comments of Reviewer 1. Below are the *original comments* followed by our response to them.

**General comments**

*The study concentrates on predisposing and triggering factors that caused a large population of landslides during the 2018 Typhoon Mangkhut, in the Philippines. The manuscript is quite clear and not difficult to read. The structure can be a bit improved, but already acceptable. Data and analyses are of some interest but there are weaknesses, in particular in the critical evaluation of the result, that makes the discussion in some parts superficial and ambiguous, also the literature review in the introduction can be improved with some more references better linked to the topic.*

The literature review has been improved (introduction and methods) as well as more details and further insights have been provided in the discussion.

Some examples of the improvement of the introduction and methods sections are:

- **More references and examples of techniques on automatic tools for landslide mapping have been included**
  References:
  (Alvioli et al., 2018; Borghuis et al., 2007; Kirschbaum and Stanley, 2018; Mondini, 2017; Prakash et al., 2020; Scheip and Wegmann, 2020)

- **More details and references on the existing hazard assessment techniques used in the Philippines and stress the improvements that our work brings**
  Methods used so far are based on heuristic approaches.
  References:
  (Aleotti and Chowdhury, 1999; Corominas et al., 2014)

- **More details and references on the uncertainties of landslide-triggering rainfalls analysis**
  For the extra details see detailed comments. New references listed here:

  (Peres et al., 2018)

  (Luigi et al., 2020)

  (Nikolopoulos et al., 2015)

- **More references on the relevance of soil moisture in the landslide triggering mechanisms**
  References:
  (Bogaard and van Asch, 2002; Rahardjo et al., 2008; von Ruette et al., 2014)

Some points that have been tackled more in depth in the discussion are:

- **The correlation between the findings in the predisposing factors and the hydrometeorological conditions that lead to the failure during Typhoon Mangkhut**

- **The comparison between the results of the rainfall analysis of this work and previous published works, despite being obtained from different sources and methods**
- **The relevance of not only using rainfall data but also soil moisture in Early Warning Systems**

You will be able to find more details of the improvements through the answers to the specific comments.

*My main concerns are:*

- *I have not found any clear connection between the meteorological information and the geo-environmental information as if they were two different topics. Is that a choice because lack of connection evidence? Are the resolutions too different for a proper comparison?*

We are thankful to the reviewer to identify that it was probably not clear enough, therefore we have stressed these links more clearly in the discussion, also emphasizing the correlation between the type of soil cover and the increase of soil moisture in the previous months to the Typhoon. We agree that resolutions are very different, but the study area is big enough to show some spatial differences for both hydrometeorological and geo-environmental factors so it is relevant to connect the two analysis.

- *Some points related to the sampling methods are unclear (to me), need for further clarifications.*

We have improved the methods section (3.3. Analysis of rainfall and soil moisture). More details can be read in the comments about "methods".

- Some of the comparisons between the results obtained in this work and elsewhere should be better contextualized, in particular when obtained with different methods or data.

We have improved the results explanation and discussion on this topic in the revised manuscript.

- Connected to the previous point, results are depending on some of the preliminary assumptions, including the definition of rain event. This is ok, but the impact of the choice should be better discussed, and can we compare products obtained with different definitions of the initial settings?

We have improved the description of the methodology regarding the definition of the rain event and made emphasis on its relevance (see comments Section 3.3).

We have clarified that rainfalls in Figure 7 are represented using duration vs. maximum **floating** intensity. We appreciate that the comparison with the other thresholds in Baguio region may have induced some confusion as this was not clearly specified.

Regarding the question of comparing results using different definitions of the initial settings, we would like to make two points:

a) The maximum **floating** intensities of the rainfall events (Figure 7) are not much affected by the duration of the rainfall event. This happens, because using this method (widely used in the calculation of IDF curves in hydrology) uses the maximum intensity in the entire rainfall event for a given floating time window, i.e.: maximum floating intensity for a 3 hours duration is the maximum intensity

for a window of 3 hours in the whole rainfall event. Therefore, the results are not affected by initiation/ending criteria of the rainfall.
For example, Figure 7:
    a. using 1 h no rainfall before and after the event

    b. using 3 hours no rainfall before and after the event

b) Certainly Figure 8 a) c) and d) are affected by the definition of the rainfall as they show implicitly or explicitly the duration of the rainfall. We have stressed this influence in the manuscript, however, we still think it is relevant to show Figure 8 results as they are an example of a potential implementation of rainfall and volumetric water content combined in an early warning system.
We appreciate this would be an interesting topic for further research.

**Specific comments**

**1 Introduction**

- *25 46% of rainfall-triggered landslides: I suggest to add 'known', probably that database is incomplete*

Thanks for the comment. We agree and we have added 'known'.

- *30 in the Philippines has still received little attention: do we know why? Just a curiosity*

We do not really know why.

- *30 results in hazard and risk assessment techniques: what techniques are the Authors referring to?*

We have added some extra information in the revised manuscript.

"…, as landslide hazard maps published earlier in the Philippines are only based on geomorphologic qualitative observations considering only worst case scenarios"

- *35 landslide inventories are often not available due to incomplete event records: not sure I understand. Are the inventory missing, or incomplete?*

Inventories are incomplete as many landslide events are not even recorded. This phrase has been rewritten to avoid confusion.

- *40 Actually the relative literature is very rich, and I don't think that this list is adequate to intercept the concept.*

We have added some extra information in the revised manuscript. We consider that going into more detail about all the available techniques is not relevant for this paper, where the final mapping was done manually.

"Therefore the use of automatic mapping tools is increasing. The current state of the art of these tools is growing, as algorithms based on different source of satellite data (visible imagery and/or radar) have been developed in the last years (Alvioli et al., 2018; Borghuis et al., 2007; Kirschbaum and Stanley, 2018; Mondini, 2017; Prakash et al., 2020; Scheip and Wegmann, 2020). These tools show great potential, especially for the application in systematic inventories after fatal rainfall events."

- *40 It is particularly challenging in regions hit by the passage of typhoons: why more challenging?*

Typhoons can make landfall to very large areas of land, triggering a high number of landslides. Manual mapping in these conditions can be particularly challenging. We have clarified this sentence.

"Manual mapping is particularly challenging in regions hit by the passage of typhoons, where the area affected by landslides can be up to hundreds of km$^2$ and the density of landslides can be very high (e.g.: Tseng et al., 2015). "

- *45 landslide magnitude distribution: is it true for all the types of landslides?*

We are not sure what the reviewer meant by this comment. It would be helpful to have some more information about what he/she means.

- *65 geographical distribution: sorry, of what? Do the authors mean the size of the study area?*

We are referring to the location and size of the of the study area, so we have clarified that in the revised manuscript.

- *70 resolution: uncertainty on the resolution. Can the Authors please unravel this concept?*

We are referring to the spatial and temporal resolution of the data. This has been clarified in the revised manuscript.

- *80: about this concept, I suggest you have a look at here.*

We are not sure where does the Reviewer exactly refer to have a look at. It would be helpful to have some more information about what he/she means.

- *85 high spatial: compared to? I actually suggest to clarify the concept, since it was told that the resolution is not enough to catch convective processes. Is the fact of working over large areas enough to consider the problem not impacting on the final results?*

Thanks for the comment. We have changed the sentence as it was confusing. We would like to stress the advantage of satellite rainfall products to have large coverage (entire globe for some satellites) at high temporal resolution. Although the high intensities of convective cells can be lower in satellite products, the fact that it is a spatial data rather than point based (as rain gauges are) is an advantage in large areas.

"A clear advantage of the satellite rainfall products is the large coverage at high temporal resolution, which enables detailed analysis of rainfall conditions that trigger multiple landslides over big regions."

- *85 In some regions of the world: it is too generic, isn't it?*

We have added some more references and rewritten this sentence as thanks to the comment of the Reviewer we noted it could be confusing as it could seem that this has only been proved some times. We consider that the influence of soil wetness in the landslide triggering mechanisms is a widely demonstrated fact, therefore it is unlikely that we could include a list of all the case studies or models where this has been proved.

"The soil wetness at the beginning of the triggering rainfall has been proven to play a major role in landslide triggering mechanisms (Bogaard and van Asch, 2002; Rahardjo et al., 2008)(von Ruette et al., 2014) and therefore to help improve the early warning systems (Guzzetti et al., 2020; Krogli et al., 2018; Marino et al., 2020)."

- *100 for the first time: do move it earlier*

We have added this earlier in the Abstract.

- *100 producing one of the first complete inventories: this sentence sounds a bit weird to me: how can more than one complete inventory exist? How do the Authors define 'complete' (and how verified: : :)*

We were referring to landslide inventories from other typhoons, as one inventory corresponds to one typhoon. We have performed some extra validation of the inventory by checking the first pre- and post- satellite images available in a narrower window to verify that the mapped landslides are certainly from Typhoon Mangkhut and not other possible rainfalls. Clearly there is some limitation on the satellite resolution and some landslides could have been missed. Therefore we have avoided the word 'complete'.

**2 Study Area**

2.1 Geological and geomorphological Settings

- *110 steep slopes: I suggest to provide here some quantitative info about the slope*

We have added more details about the slope.

"The valleys are characterized by steep slopes (30 degrees on average, but up to 70 degrees),..."

- *120 The study area can also be described as seismically active: according to: : :.*

A reference has been included.

Su, S. S.: Seismic hazard analysis for the Philippines, Nat. Hazards, 1(1), 27–44, doi:10.1007/BF00168220, 1988.

2.2 Climate

- *2.3 Landslide activity: not sure I would use 'activity' here*

We agree maybe it could lead to misunderstandings with the landslide frequency. Therefore we have rewritten the title of the section as:

"Landslides related to previous typhoons"

- *140 simultaneous in 2009: overlapping or in two different parts?*

Trajectories of Super Typhoon Parma and Super Typhoon Melor interacted on October 6 2009 and Parma was drawn towards Melor. They did not overlap, we have clarified this in the revised manuscript.

"...Typhoon Parma (influenced by simultaneous Tyhoon Melor in 2009), with 97 landslides reported..."

- *150 The rapid urbanization, together with the ongoing mining activity also represents a relevant factor in landslide risk in the area, though these factors were not considered in this study (Mines and Geosciences Bureau, 2018). : this sentence, here, is out of context.*

As it was also mentioned in the discussion, we have deleted this from this section of the paper.

2.4. Typhoon Mangkhut (13-15 September 2018)
- *155: The highest observed 4 day rainfall total (12 to 15 September 2018) of 794 mm in Baguio City PAGASA weather station was recorded due to the passage of Typhoon Mangkhut (Weather Division PAGASA, 2018).: I suggest to re-phrase*

We have rewritten this part.

"The highest rainfall amount recorded during the passage of Mangkhut was at Baguio City PAGASA weather station (at the West of the study area), and it was 794 mm from 12 to 15 of September (Weather Division PAGASA, 2018). "

**3 Data and methods**

3.1 Compiling a landslide inventory and magnitude-frequency analysis

- *170We experimented with an automatic landslide mapping tool to map landslides more efficiently: this sentence is too generic and probably in the wrong place (maybe discussion?).What tool? Set up? How did the Author manage the different images? What does the success rate was low mean? And compared to what?*

We moved this topic to the discussion. We have also added some more information about the tests we did with the automatic tool and highlighted the existence and recent developments of similar tools (in the introduction). However, the scope of this paper was not the comparison of automatic tools, therefore we don't think it is relevant here to go into further discussion about the topic. Further details on the work we carried out can be found in:

*Abancó, C., Bennett, G., Briant, J., and Battiston, S.: Towards an automatic landslide mapping tool based on satellite imagery and geomorphological parameters. A study of the Itogon area (Philippines) after Typhoon Mangkhut, EGU General Assembly 2020, Online, 4-8 May 2020, EGU2020-17940, https://doi.org/10.5194/egusphere-egu2020-17940, 2020*

Finally, we would like to mention that we are working on the test and validation of automatic techniques, not only in this area but other areas of the Philippines in the context of SCaRP project.

The text included in the discussion of the reviewed manuscript is:

"This study has been conducted using a manually-mapped landslide inventory. We experimented with an automatic landslide mapping tool to map landslides more efficiently, based on the application of a threshold for NDVI using a random forest model and a post-process by filtering flat areas (Martinis, 2018).When comparing with visual observations, we found the success rate insufficient, as the tool was only designed for the use Sentinel-2 images and it was unable to detect smaller landslides  (<800 m$^2$) (Abancó et al., 2020). Despite the potential of other types of automatic tools, for this specific work the final inventory was entirely done using manual techniques, combining very high and high resolution imagery in order to narrow down the time windows and ensure we were mapping landslides triggered by Typhoon Mangkhut. "

- *175 limitations: or problems? I also suggest to refer to optical remote sensing here,SAR is starting to provide some alternatives, also for such large events. I suggest to have a look at:Measures of spatial autocorrelation changes in multitemporal SAR images for event landslides detection (2017) AC Mondini. Remote Sensing 9 (6), 554.*

As mentioned in the General comments, some more references have been added in the introduction, and this is one of them.

- *180 have been used to cross-check part of the inventory: and the result is...*

Planet Labs imagery pre (06/09/2018) and post (19/09/2018) typhoon narrower than in the 1$^{st}$ version of the manuscript) was used to verify that the landslides included in the inventory

did happen during the passage of Typhoon Mangkhut. We have added this in the revised manuscript:

"For this reason, other imagery sources with narrower time windows (only few days in the case of Planet Labs) (Table 2) and Google Earth, together with the comparison with local reports reporting field surveys after the Typhoon (Mines and Geosciences Bureau, 2018) have been used to cross-check the inventory and verify that the landslides mapped did actually occur during the passage of Typhoon Mangkhut."

**3.2  Analysis of landscape controls on landslides**

- *200 governing: I suggest to change the term*

We have changed governing for predisposing factor.

**3.3 Analysis of rainfall and soil moisture**

- *215 total rainfall (from the start of the rainfall until it finishes): It sounds contradictory with "These data are of particular interest to identify any correlation between the spatial variability of the rainfall associated with Typhoon Mangkhut and the distribution of landslides, instead of having only the point-based data from Baguio city rain gauge". I suggest to better specify that you are doing a zoom in.*

We have rewritten this section clarifying that the analysis of the rainfall had two main purposes: a) to analyse the spatial distribution of the triggering and antecedent rainfall over the study area and b) to zoom into one grid point (the nearest to the fatal landslide at Barangay Ucab) and perform a more detailed analysis of rainfall and soil moisture, and compare with previous results in the area and worldwide.:

"These data are of particular interest to  analyse: a) the  correlation between the spatial variability of the rainfall associated with Typhoon Mangkhut and its antecedent rainfall and the distribution of landslides (instead of having only the point-based data from Baguio city rain gauge (**¡Error! No se encuentra el origen de la referencia.**)); and b) the characteristics of the Typhoon Magkhut rainfall and soil moisture at the nearest GPM grid point to the fatal landslide in Barangay Ucab."

We have also clarified what the analysis near Barangay Ucab was performed only in one GPM grid point, following the suggestion of RC1:

"We considered that the  total rainfall of the event was that which occurred between the beginning and the end of the rainfall at the nearest GPM grid point to Barangay Ucab."

- *220 total rainfall (from the start of the rainfall until it finishes): I suggest to define here what start and end mean*

We have rewritten most of this section, and now the definition of the beginning and end of the event are earlier in the text:

"We considered that the total rainfall of the event was that which occurred between the beginning and the end of the rainfall at the nearest GPM grid point to Barangay Ucab. We assumed that a rainfall event starts and ends after and before a period of 1 hour of no rain, following Abancó et al. (2016). "

- *225 did not trigger landslides: this is a critical point, how to make sure that landslides did not occur? Please, do see a previous comment on the inventory check.*

Please see comments in section 3.1.

- *225-230 relevant: the previous numbers are objective, this sentence is relevant. I suggest to explain better why this is relevant. Furthermore, can this rule be applied to all the other events? Perhaps rules should change according to the type of rain event: : :*

We agree that this is a very relevant point of the landslide triggering rainfall analysis, therefore we have extended the explanation regarding this aspect:

"The definition of the rainfall duration is a key consideration in the analysis of the rainfall thresholds for landslides, which often brings uncertainty to the analysis (Abancó et al., 2016; Luigi et al., 2020). Frequently the information of the failure time of landslides is unknown, hence discriminating between the rainfall that occurred before and after the failure becomes challenging. The assumption that the amount of triggering rainfall is actually the precipitation from the entire rainfall event (from the beginning until it finishes, in the day of the landslide event) is common in the literature, however it has been proved that if the uncertainty of the landslide occurrence spans more than one day triggering rainfall? can be significantly underestimated (Peres et al., 2018).  "

We have also improved the explanation about the selection of high intensity rainfalls.

"In order to select high intensity rainfall events we filtered rainfalls with intensity higher than 4 mm hr-1 in average for 3 consecutive hours, which would mean at least an accumulated rainfall of 12 mm in 3 hours. Although 12 mm may not seem a high amount of rainfall, the selection criteria was based on the fact that only 3% of the 30 minute rainfall records from GPM IMERG exceeded 4 mm hr-1 in 2018. The mean daily rainfall of 2018 was 9 mm day-1, and only 34 rainfall events fulfilled the condition of having an intensity of 4 mm hr-1 for 3 consecutive hours at the grid point near Barangay Ucab. "

Regarding the comment about the different rules for different types of rain events, we consider that this would be an interesting topic to analyse in a future analysis but is beyond the scope of this paper.

**4 Results**

4.1 Landslide characteristics

General questions:

- *1) for every single landslide, all the pixels were used, or only one 'representative'?*

We used the mean value for each single landslide, we have clarified this in the ms.

- *2) are the pixels inside and outside landslides comparable in number? Normalized?*

They are not comparable in number, because area affected by landslides only represents less than 1% of the study area. For this reason, the normalized frequency distribution (#landslides/km2) was included in Figure 4.

The combination of the histograms of:

- Only areas affected by landslides (mean values for each class)
- Number of landslides/km2 of study area

Provides the general idea of how the areas are affected by landslides and how they are in respect to the entire study area.

- *245 figure 3: actually difficult to see the rollover. I recommend to add information about the quality of the fit (uncertainty)*

We have added some extra information in the revised manuscript regarding the methodology used to fit the power-law distribution. The rollover may be difficult to see due to the methodology we used. We have used the complementary cumulative distribution function (CCDF) following Clauset et al. (2009), which is obtained by integrating the following Equation:

$$P(x) = P(X \geq x) = \left(\frac{x}{x_{min}}\right)^{-\rho}$$

Where $P(x)$ is the probability of a randomly picked failure area exceeding x and $\rho$ is the slope of the CCDF.

The use of the CCDF is thought to be preferable to the standard PDF (Probability distribution function) as it avoids the ambiguities introduced by arbitrary selection of bin sizes or scale, however, the rollover is less easily visualized (Bennett et al., 2012).

- *245 maximum in elevations: the peak of the normal distribution?*

We have changed maximum by peak.

4.2 Rainfall and soil moisture conditioning and triggering of landslides

    4.2.1 Rainfall
- *275 timing: time of occurrence?*

Yes, we have clarified this.

    4.2.2 Soil moisture
**5 Discussion**

- *315 Using a landslide inventory based on a single event provides information that is strongly influenced by the event characteristics itself: I suggest to re-phrase.*

We have rewritten this as: "This study has been conducted using a landslide inventory based on a single typhoon, therefore the results may be conditioned by the characteristics of this specific event."

5.1. Landslide characteristics and landscape preconditioning

- *325 we fill this gap in the literature here: it is actually an example, I would be a bit less 'absolute': : :.*

We have rewritten this as: "We present here a magnitude-frequency distribution of landslides, which is, up to the knowledge of the authors, the first published one in this area of the Philippines"

- *330 small landslides are more frequent than larger ones: in fact, it is difficult to see the rollover. I think this comparison is interesting, but it is missing a few of elements: type of used data, mapping methods, fitting procedures (the constrains imposed by the fitting), and others should be better unraveled.*

We have commented on this in Section 4.1. The type of data used were the landslide areas, not including the runout for the landslides where the two components (failure/runout) can be distinguished.

- *330 Further mapping in the region and across other regions of the Philippines will help to refine these distributions and exponents: this sentence sounds a bit weird. The distribution here is related to this event and in the local geo-settings, while the sentence seems to look for a general behavior.*

We have rewritten this as: "Further mapping in the region and across other regions of the Philippines will improve the general knowledge of the frequency of occurrence of landslides and their magnitude in the country."

- *335 aspect: how about anaclinal, cataclinal??*

Information on bedrock dip direction was unfortunately not available. The thickness of soils and highly fractured rocks is up to 20 meters in the area (Nolasco-Javier and Kumar, 2018), while some of the failures are much superficial (<10 m) therefore we think it is unlikely that this has a strong influence. However, we will consider this for further research in the area.

We have included further analysis regarding the winds during the Typhoon. Winds during the highest intensity rainfall were coming from West-Southwest, which does not reveal a clear explanation for the aspect control. For this reason further research, including anthropogenic factors (see below) is needed.

- *360: how local are these effects?*

Excavation of underground mines in the area has been done without clear regulation. At the moment, the authorities don't have a record or map of the adits, therefore it is difficult to determine how much influence this factor has. Mainly for this reason this factor has not been included in this analysis, although its important has been pointed out by some local project partners and we are considering gathering data and analysing it in further steps of the research.

5.2. Rainfall and soil moisture conditions leading to landsliding

- *365-380: I have some concerns about this sub-paragraph because it is unclear to me whether the different results can be really compared since the definition of the events are different, but more critical, the data are eventually different!!*

We thank the reviewer to point out that this part was not clear enough and probably that is why the reviewer was confused about the comparison of different results. We have rewritten this section, and the main points are:

1) The spatial analysis of the rainfall distribution shows that landslides didn't occur mostly in areas of high rainfall intensity but where more antecedent rainfall happened.
2) If we zoom in to one GPM point (near Barangay Ucab) and we compare the annual rainfall timeseries we find out that the threshold suggested by Nolasco-Javier and Kumar (2018) (500 mm of cumulated rainfall in the rainy season) was exceeded much earlier than where the landslides happened. It would be useful to have longer timeseries to refine it; defining rainfall thresholds is a challenging task.

3) If we zoom in to one SMAP4 point, we can see that the the soil moisture values at the beginning of Typhoon Mangkhut are likely to be the saturation point of Bakakeng clays (where most landslides occurred). This highlights the potential of using Early Warning Systems combining both parameters: rainfall and soil moisture.

- *385 This value is actually a reasonable value for the porosity of clays,: perhaps, but I think this should be supported by evidence, papers, references..*

We have included a reference:

"Hough, B.: Basic soil engineering, edited by R. P. Company, New York., 1969"

5.3. Potential of satellite-based rainfall and soil moisture data for landslide early warning

- *395 Alternatively, it may be that satellite-based rainfall and soil moisture data do not adequately: see one of my previous comments. Not adequate, or not comparable..*

We agree with the reviewer that rainfall and soil moisture and rainfall are actually different factors involved in the triggering mechanism of shallow landslides and they cannot be compared (see comment in previous section for more details). However, in this last part of the analysis, we are not trying to compare these two factors but to see if a combination of both of them would be useful for the implementation of an Early Warning System. We have clarified this in the revised manuscript.

"The purpose of this analysis was to find out if by combining information on: a) the initial soil moisture at the beginning of a rainfall event and b) the characteristics of the rainfall, it would be possible to discriminate between critical and non-critical rainfalls."

---

## Author Comment (AC2) · 17 Dec 2020

**Response to Xiangzhou Xu (RC2)**

We are grateful for the detailed reading of the paper and substantive comments of Reviewer 2 (Xiangzhou Xu). Below are the *original comments* followed by our response to them.

**General comments**

*Landslide plays an important role in landscape evolution, delivers huge amounts of sediment to rivers and seriously affects the structure and function of ecosystems and society. This paper, which is entitled "The role of geomorphology, rainfall and soil moisture in the occurrence of landslides triggered by 2018 Typhoon Mangkhut in the Philippines", tries to examine the factors susceptible to landslides, consider the potential for early warning of the landslides. The topic looks very interesting and valuable. Nevertheless, a major revision is needed before the manuscript is accepted for publication in the journal NHESS.*

**Specific comments**

*Some problems are listed as follows:*

1. *Title: I suggest you erase the words "triggered by 2018 Typhoon Mangkhut" in the title. I think the readers will be interested in a relatively universal law related to landslides instead of only a certain storm. Also you have to add an in-depth discussion corresponding to the title revision.*

We consider that due to the specificity of the study, referred to Typhoon Mangkhut, it is important to state this in the title. Although it is a site and event specific study, it provides key data and results that, together with further analysis, will help stating general laws about the triggering mechanisms of landslides in the Philippines. Throughout the paper we mention several times the importance of repeating similar analyses for other areas and meteorological events in order to generalise some conclusions.

2. *Abstract: This part has to be rewritten. The point, "a) it was one of the most intense rainfall 20 events in the year but not the highest", is a condition to induce the landslides instead of a result. In addition, have you resolved the problems presented in lines 99-104 (page 4), Section Introduction? Please let me know in the abstract with a concise description.*

We have rewritten the abstract following the suggestions of the reviewer.

"In 2018 Typhoon Mangkhut (locally known as Typhoon Ompong) triggered thousands of landslides in the Itogon region of the Philippines. A landslide inventory of Typhoon Mangkhut is compiled for the first time, comprising 1101 landslides over a 570 km2 area. The inventory is used to study the geomorphological characteristics and land cover more prone to landsliding as well as the hydrometeorological conditions that led to widespread failure. The results showed that landslides mostly occurred in slopes, covered by wooded grassland in clayey materials predominantly facing East-Southeast. Rainfall (GPM IMERG) associated with Typhoon Mangkhut is compared with 33 high intensity rainfall events that did not trigger regional landslide events in 2018. Results show that landslides occurred during high intensity rainfall, coinciding with the highest soil moisture values (clays saturation point), according to SMAP-L4 data. This indicates that, in addition to rainfall from the typhoon, soil moisture plays an important role in the triggering mechanism. Our results suggest that SMAP-L4 and GPM IMERG data show potential for landslide hazard assessment and early warning where ground-based data is scarce. However, other rainfall events the months leading up

to Typhoon Mangkhut that had similar or higher intensities and also occurred when soils were saturated did not trigger widespread landsliding, highlighting the need for further research into the conditions that trigger landslides in typhoons."

3. *Study area. Too many details have been given in Section 2. You may delete some descriptions which are not closely related to the topic of the paper, and the subtitles of the section, including subtitles 2.1-2.4.*

We appreciate the comment, however we think all the information included in Section 2 is relevant for the understanding of the different methods and results of the article. We have rewritten some sentences in this section, as per Reviewer 1 comments for clarification.

**4. Methods.**

*(1) Line 110, page 4: What's the meaning of the unit "m.a.s.l"?*

m.a.s.l. means "meters above sea level", we have clarified this.

*(2) How to distinguish a landslide in the area with scarce plants?*

We actually decided to do our mapping manually instead of using automatic tools, which normally rely on fewer parameters, such as vegetation change (NDVI). Changes in vegetation were really helpful in the area to identify landslides, but they were not the only ones. High resolution imagery was used to identify landslides also based on changes in sediment accumulation in the lower part of the slopes or along roads.

An example:

a)                                                b)

[Figure]

[Figure]

c)                                                d)

[Figure]

[Figure]

Figure 1: Pre Mangkhut (a and c) and post (b and d) images from Baguio airport runway landslide. Images a) and b) are an ortophoto, c) and d) are Google Earth images (note that North is at the bottom right of the picture).

*(3) I do not think a rainfall with the intensity of 4 mm hr-1 is intensive rainfall. Line 374, page 12. You said "Our study suggests a threshold of 2600 mm of rainfall accumulated over the rainy season for landsliding to occur". However, in lines 226-227, "Our selection of high intensity rainfall events was. . . exceeding 4 mm hr-1". The intensity of 4 mm hr-1 is really too small. Maybe 4 mm min-1?*

We appreciate that the explanation of the selection criteria for high intensity rainfalls was confusing. Therefore, we have improved this explanation in the revised manuscript:

"In order to select high intensity rainfall events we filtered rainfalls with intensity higher than 4 mm hr-1 in average for 3 consecutive hours, which would mean at least an accumulated rainfall of 12 mm in 3 hours. Although 12 mm may not seem a high amount of rainfall, the selection criteria was based on the fact that only 3% of the 30 minute rainfall records from GPM IMERG exceeded 4 mm hr-1 in 2018. The mean daily rainfall of 2018 was 9 mm day-1, and only 34 rainfall events fulfilled the condition of having an intensity of 4 mm hr-1 for 3 consecutive hours at the grid point near Barangay Ucab. "

We would like to include two Figures to visualize more graphically the selection criteria for rainfalls that we considered "high intensity". Note that Figure 2a refers to the daily rainfall and Figure 2b to the rainfall intensity. In both figures the threshold for high intensity rainfalls is indicated, but in Figure 2b it must be noted that it have to last for 3 consecutive hours.

[Figure]

Figure 2: a) Daily rainfall and b) rainfall intensity along 2018 at the nearest GPM point of Barangay Ucab.

**5. Discussion.**

*Section 5.1. What's the meaning of preconditioning, and have you discussed the preconditioning here?*

We have actually changed the concept preconditioning for predisposing.

**6. Figures and tables.**

*(1) Most of the figures are not clear. The sizes of the texts in some figures are too small. The figures should be clear as they are printed in black and white.*

We have increased the font size and changed changed the markers in order to make it evident for b/w prints of Figure 9

We have increased the font size of Figure 8 and Figure 4

We think the colours and sizes of all the Figures are correct now.

*(2) Figure 1. What are the differences between the "study area" and "Regions Study Area"?*

In section 2.1 the text is referring to different Regions in the study area with different geomorphologic features. We have added the reference to the section in the Caption of Figure 1 of the revised manuscript

*(3) Figures 2 and 8. The general titles of the figures are needed.*

According to the guidelines of NHESS for figures and tables, the description of the figure must be added in the caption but not in a title. We consider that the captions are explicative enough to not need general titles in the figures.

*(4) Figures 9 and 10. The scales of the vertical coordinates are anticipated.*

We are not sure what the reviewer tries to point out here. It would be helpful to have some more information about what he means.

*(5) The format of the table in the manuscript is not suitable to the requirements of the journal NHESS.*

We have changed the format of both tables.

*7. The English writing of this paper is readable. Nevertheless, still some minor language errors exist, e.g., the word "are" in line 153 of page 5 should be erased; in line 51 of page 2, the words ", however the. . ." may be replaced with the words "; however, the. . ."; the first letter of the word "Clay" in line 256 of page 9, may be in lower case.*

We appreciate the comments:

- "are" has been erased
- "however the" a comma has been added
- "Clay" upper case has been substituted by a lower case

---

## Author Response (AR2)

**Response to editor**

We are grateful for the positive feedback from the two reviewers. We have checked and improved the minor issues that they raised. The changes made to the manuscript are listed in the following:

- An exhaustive English language revision has been done
- Grey lines of text in the Figures have been changed to black and symbols indicating the digits have been added.

Regarding the standing question of Reviewer 1:

*45: magnitude frequency analysis: here magnitude = area, but in general magnitude should refer to energy, so I guess that the magnitude is proportional to the area of the landslide (or a proxy), then my question was: is this reasonable for all the different types of landslides, e.g. rock falls?*

The magnitude of landslides is commonly represented as the volume of the moving mass. In the case of shallow landslides it is common to use the area instead of the volume (e.g.: Guthrie and Evans, 2004, 2005), is the third dimension is clearly smaller (shallow). For rockfalls the magnitude is generally represented as the volume of rock mass.